# Graph Concept Bottleneck Models

## Abstract

Concept Bottleneck Models (CBMs) provide explicit interpretations for deep neural networks through concepts and allow intervention with concepts to adjust final predictions. Existing CBMs assume concepts are conditionally independent given labels and isolated from each other, ignoring the hidden relationships among concepts. However, the set of concepts in CBMs often has an intrinsic structure where concepts are generally correlated: changing one concept will inherently impact its related concepts. To mitigate this limitation, we propose **Graph CBMs**: a new variant of CBM that facilitates concept relationships by constructing latent concept graphs, which can be combined with CBMs to enhance model performance while retaining their interpretability. Empirical results on real-world image classification tasks demonstrate Graph CBMs are (1) superior in image classification tasks while providing more concept structure information for interpretability; (2) able to utilize concept graphs for more effective interventions; and (3) robust across different training and architecture settings.

## 1 Introduction

Deep neural networks have shown supremacy and efficiency in various tasks in Computer Vision He et al. (2016); Dosovitskiy et al. (2021); Tolstikhin et al. (2021), Natural Language Processing Vaswani et al. (2017); Devlin et al. (2019); Brown et al. (2020); Raffel et al. (2020), and Graphs Kipf & Welling (2017); You et al. (2020b); Rong et al. (2020b). Despite their great performance and expressivity, most deep learning models are often regarded as black-box models since they lack interpretability. However, in areas such as medical applications, there is a surging need to explain the prediction of deep learning models to make them more transparent and reliable.

To gain better interpretability, Concept Bottlenectk Models (CBMs) Koh et al. (2020) is proposed to map inputs' hidden representations to a concept score space, in which each hidden neuron in the concept bottleneck layer corresponds to an interpretable concept with a confidence score. Unlike End-to-End training that maps the input (e.g. pixel values in images) directly to target labels, CBMs make a prediction based on the representations inside concept space. In this manner, we can tackle the black-box problem and intervene in concept score vectors to adjust the final prediction without changing the model's parameters.

However, limitations still exist inside current CBMs. One major limitation is missing correlations among concepts: the existence of a group of concepts can reinforce or diminish other concept scales. For example, when one is prompted with concepts like 'furs' and 'whiskers', one may anticipate concepts like 'tails' or 'paws' shown up; while if concepts were 'wings' and 'beaks', then 'feather' would be more possibly present. Previous methods Espinosa Zarlenga et al. (2022); Kim et al. (2023); Yuksekgonul et al. (2023) have proposed many variants of CBMs to have a more sophisticated module to generate concept scores. However, most still ignore the structure and interactions of the latent concept.

Motivated by a lack of internal concept structure, in this paper, we aim to capture the hidden interactions within the concept space while preserving the essence of CBMs. We propose **Graph Concept Bottleneck Models (Graph CBM)** which uses a latent concept graph to explicitly enhance the concept interaction and update concept scores by aggregating information from its neighborhood. Observing that the activated concept graph can be regarded as another view of an image, we design a contrastive learning mechanism to learn the latent graph; updated concept scores are then fed into a classification module for final prediction. Based on different backbones for the classification module,

we developed four variants: 1) Graph LF-CBM and Graph PCBM for label-free settings; 2) Graph CBM and Graph CEM for concept-supervised settings.

To summarize our contribution:

- We propose Graph CBMs that utilize a learnable graph structure to allow interactions among concepts. Empirical results have shown their effectiveness in both target prediction and intervention when compared to SOTAs.
- We design a contrastive learning framework to automatically learn the latent concept graph. Our Graph CBMs have demonstrated robust performance across different settings and backbones, and Graph CBMs can still be effective when there are concept interventions.

## 2 RELATED WORK

**Concept Bottleneck Models**: Koh et al. (2020) first presents the idea of Concept Bottleneck Models, tackling the black-box property in the End-to-End training paradigm by mapping inputs to certain concepts and then classifying concept vectors. Espinosa Zarlenga et al. (2022) further investigates CBMs and proposes a new architecture called Concept Embedding Models (CEM). Kim et al. (2023) continue to work on the CEMs by modeling probabilistic embeddings in the concept embedding space and the class embedding space. Havasi et al. (2022) addressed the issues of insufficient concept set and inexpressive predictor. Xu et al. (2024) applies energy-based models to define joint energy of (input, concept, class) tuples. While these works show some promising results, they require collecting concept labels to train the models, which may be expensive and time-consuming with manual specifications. To address this shortcoming, recent works Oikarinen et al. (2023); Yang et al. (2023) leverage large pretrained models to enable automation.

Yang et al. (2023) leverages a language model to define a large space of possible bottlenecks and filters the concept set with a submodular utility that selects discriminative and diverse information. Yuksekgonul et al. (2023) introduces Post-hoc Concept Bottleneck models (PCBM) which mitigate the requirement of dense concept annotations and avoid training a backbone from scratch by only training the last FC layer along with an optional residual fitting layer. Label-free CBMs (LF-CBM) Oikarinen et al. (2023) transform any neural network into an interpretable CBM without labeled concept data and use a sparse prediction layer to enhance model accuracy and interpretability simultaneously. On the other hand, the above methodologies ignore the interaction between concepts during inference and intervention, our architecture is orthogonal to them and can be integrated with them efficiently.

**Graph Structure Learning**: Graph Structure Learning (GSL) is a standing problem that finds useful graph structures from data. It has been explored in different contexts and domains. Early efforts come from the signal processing community Dong et al. (2019). Another field of study is grounded by probabilistic graphical models. For example, Yu et al. (2019) intends to learn a faithful directed acyclic graph (DAG) from samples of a joint distribution and propose DAG-GNN that naturally handles discrete variables and vector-valued ones. In the era of graph neural networks, the latent graph is often learned together with the graph neural network parameters Franceschi et al. (2019); Kipf et al. (2018); Shang et al. (2020); Ma et al. (2023); Kazi et al. (2022). For example, NRI Kipf et al. (2018) adopts a latent variable approach to learn the latent interaction graph of dynamic systems; Franceschi et al. (2019) formulates graph learning as a bi-level optimization problem, and learns discrete probability distribution on the edges of the graph. Wei et al. (2022) construct contrastive views for HyperGraphs via studying graph structures in the HyperGraph setting. In our case, we define a learnable deterministic graph structure that can be trained together with graph neural networks by contrastive learning.

## 3 METHOD

### 3.1 PRELIMINARY

**Concept Bottleneck Models**: Given a set of image $V = \{v_1, v_2, \ldots, v_n\}$ and a set of concepts $T = \{t_1, t_2, \ldots, t_k\}$. Each $v_i$ is associated with a label $y_i \in \mathcal{Y}$. The model, CBM, consists of two mappings: $f_1$ maps image space to the concept score space, and $f_2$ maps the concept space to the label space. To predict the image label, the CBMs first project the input images to the concept space

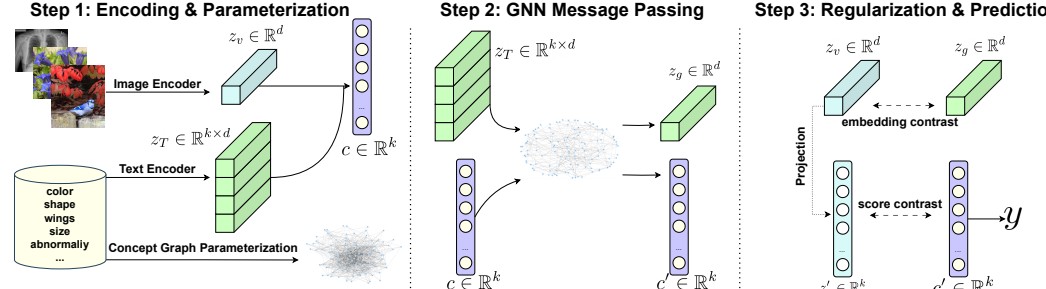

Figure 1: Overview of Graph CBM. Step 1: Compute our initial concept score vector by measuring the distance between image embeddings and text/concept embeddings, and parameterize a learnable latent graph according to the concept set. Step 2: Update the concept embeddings and concept score vectors through message passing. Step 3: Use different granularity contrastive losses to qualify the latent graph (as the dense graph in step 1 to the sparse graph in step 2) and make final predictions.

to get the concept scores using a function $c_i = f_1(v_i; T)$, and then use concept scores to predict the final label $\hat{y}_i = f_2(f_1(v_i; T))$. We define $\forall 1 \leq i \leq n : c_i \in \mathbb{R}^k$, i.e., $c_i$ will be a concept score vector and each entry value $c_i^j$ ($1 \leq j \leq k$) tells the correspondence between image $v_i$ and concept $t_j$.

**Label-Free CBMs**: The original CBMs require concept annotations for supervised training of function $g$, however in most cases, there are no annotated concepts. One common way to solve this problem (as shown in Oikarinen et al. (2023); Yang et al. (2023); Yuksekgonul et al. (2023)) is to find related concepts from external knowledge and use a pretrained multi-modal encoder (e.g. CLIP Radford et al. (2021)) to assist the concept prediction. The concept scores can be directly used for prediction Yang et al. (2023), while it can also be used to guide another backbone image encoder and obtain a concept bottleneck layer (as in LF-CBM Oikarinen et al. (2023)).

**Our Setting**: Given an image $v_i$ and a concept set $T$, our goal is to find a combination of activated concepts that delineates the image well and can be classified into the corresponding label class. Considering the lack of ground true concept annotations in most real-world datasets, we follow the settings of label-free models such as LF-CBM Oikarinen et al. (2023), and assume to have pretrained multi-modal encoders $E_v : V \to \mathbb{R}^d$ and $E_t : T \to \mathbb{R}^d$ that map images and concepts into a shared hidden space, where the image hidden feature $z_v = E_v(v) \in \mathbb{R}^d$ is well-aligned with the concept/text feature $z_t = E_t(t) \in \mathbb{R}^d$. We denote $z_T = E_t(T) \in \mathbb{R}^{k \times d}$ to be the hidden features of the entire concept set. For each image $v_i$, the initial concept score vector can be calculated through the dot product: $c_i = z_{v_i} z_T^\top \in \mathbb{R}^k$.

## 3.2 GRAPH CONCEPT BOTTLENECK MODELS

The main motivation of the Graph CBM is to bring latent graph structures to the concept space. Latent graphs exist in various scenarios and inferring the latent graph topology (or called graph structure learning) has been demonstrated beneficial in different applications, such as time series Shang et al. (2020), physical systems Kipf et al. (2018), computer vision Kazi et al. (2022), and it is even useful when the data already has a graph structure Franceschi et al. (2019). In the context of concept bottleneck models, it is clear that many of the concepts for image classification have intrinsic interactions, and the concept scores are influenced by each other. A concept graph captures such interactions by viewing concepts as nodes and connecting related pairs through edges. The natural interpretability of graphs also provides another perspective to explain how the model learns to make predictions through concepts.

To learn the latent graph, we employ a contrastive learning framework, with fundamental assumptions that 1) the structure of the concept graph shall be label-agnostic; 2) each image can be seen as a combination of a set of activated concepts in the concept graph. Thus, the activated concept graph provides an augmented view of input images, and the representation from the activated concept graph and the image encoder should have the maximum mutual information. Once the latent graph is learned, it can be used to update the concept scores and project them to the label space for classification. The framework is shown in figure 1.

We assume the latent graph among the concepts has a learnable non-negative symmetric adjacency matrix $\mathcal{A}$. With the graph, the concept scores can be updated by receiving the "message" from its neighborhood using either a graph neural network or a simple graph propagation. For simplicity, we utilize a non-parameterized graph propagation mechanism (Zhou et al. (2003); Gasteiger et al. (2019)) to update the concept score $c_i' = (1 - \lambda)(I - \lambda\hat{\mathcal{A}})^{-1}c_i$ where $\hat{\mathcal{A}}$ is a normalized adjacency matrix derived from $\mathcal{A}$, and $\lambda$ is a hyperparameter in $[0, 1)$ [1]. The inverse matrix is hard to compute and may bring in unrelated long-range dependency noise, we approximate the propagation following APPNP Gasteiger et al. (2019) and we only take its one-step approximation.

$$c_i' = f(c; \mathcal{A}) = \lambda\hat{\mathcal{A}}c_i + (1 - \lambda)c_i \tag{1}$$

$$\hat{\mathcal{A}} = I + D^{-\frac{1}{2}}\mathcal{A}D^{-\frac{1}{2}}, \tag{2}$$

where $I$ is the identity matrix and $D$ is the degree matrix. Analogous to the original CBM, we can then establish a final classification layer $h(\cdot) : \mathbb{R}^k \to \mathcal{Y}$ to predict the final label.

### 3.2.1 Contrastive Learning for Graph Structure Learning

Graph structure learning is often in a self-supervised Kipf et al. (2018) or a semi-supervised learning manner Franceschi et al. (2019). As an intrinsic structure, we expect that the concept graph should not depend too much on the labels; otherwise, the model can easily learn shallow and superficial dependencies among concepts and labels because the objective function acts on a Cartesian space of concept and label. Thus, having a label-agnostic means to learn the latent graph is crucial. We intend to discover the hidden correlations inside concept space solely, so we propose a contrastive learning mechanism to self-supervisedly learn the graph. The essential assumption behind CBMs is to view images as a combination of concepts; in other words, the concept graph representation and the image representation for the same image should be regarded as a positive pair in contrastive learning.

Given a concept graph $g_i$ for each image $v_i$, each node $j$ of the graph is a concept, with the text description $t_j$ and a concept score $c_i^j$. Although the graph structure $\mathcal{A}$ is unified at the dataset level and fixed for all images, the concept scores decide a different subset of concepts that are activated by each image. That means for each image we can have a graph with different node activations. Thus, the graph representation should be an augmented view of the image representation $z_{v_i}$.

To encode this concept graph $g_i$, we use a graph neural network (GNN) and integrate concept scores and descriptions in the nodes. In detail, we first use the text encoder $E_t$ to get the concept embeddings $z_T = E_t(T)$. After getting the concept score vector $c_i$, we will use it to weight the concept embedding for node initialization in the GNN: $z_{T,i} = c_i z_T$. The $z_{T,i}$ reflects the concept description and activation information. We feed $z_{T,i}$ into the GNN to obtain the graph embedding. To prevent overfitting and over-smoothing, we employ the DropEdge Rong et al. (2020a) to randomly drop out some edges and sample a subgraph in each layer of the GNN. For simplicity, at each layer, we use the graph convolutional network (GCN) Kipf & Welling (2017) to update the node representation:

$$z_{T,i}^l = \sigma(c_i \hat{A}_l \, z_{T,i}^{l-1} \, W_l), \tag{3}$$

$$z_{g_i} = \text{Readout}\,(c_i\text{GCN}(z_T, c_i; \mathcal{A}, \theta))$$

where $\hat{A}_l = D_l^{\frac{1}{2}} A_l D_l^{\frac{1}{2}}$, $D_l$ is the degree matrix of $A_l$, $A_l$ is the $l$-th edge-droped graph from $\mathcal{A}$, $W_l$ is the learnable weight matrix, and the superscript $l$ indicates the current index of the GCN layer or the sampled learnable subgraph; in practice, the number of sampling and GCN layers is set to be 3.

We perform a weighted readout function at the end to get the concept graph embedding $z_{g_i} \in \mathbb{R}^d$, where the weights are concept scores $c_i$. Since this graph embedding is derived from the concept embeddings $z_T$, due to the multi-modal encoder, it is still in the same space as the image embedding $z_{v_i} = E_v(v_i)$.

Thus, given a batch of $n$ images, with both the concept graph representation $z_{g_i}$ and the image representation $z_{v_i}$, we define a contrastive learning Chen et al. (2020) loss based on the normalized

---

[1]In practice, we fix $\lambda$ at 0.9 following Gasteiger et al. (2019)

temperature-scaled cross-entropy loss (NT-Xent) Chen et al. (2020):

$$\mathcal{L}_{emb} = -\log\left(\sum_{i=1}^{n} \frac{e^{\text{sim}(z_{v_i}, z_{g_i})/\tau}}{\sum_{j=1, j\neq i}^{n} e^{\text{sim}(z_{v_i}, z_{g_j})/\tau}}\right), \tag{4}$$

$\text{sim}(\cdot, \cdot)$ is the cosine similarity metric, and $\tau$ is the temperature hyperparameter (we set $\tau = 0.3$ in practice). In a mini-batch, the positive pair is the corresponding concept embedding combination, i.e., the graph feature $z_{g_i}$, and the negative pairs are those graph features of the other images.

However, only considering $\mathcal{L}_{emb}$ yields a suboptimal performance, and we have shown it empirically in Tables 9 and 10. Motivated by Ribeiro et al. (2017); Sun et al. (2019); You et al. (2020a); Wang et al. (2022), we design a second contrastive loss at different granularity to further regulate the updated concept score $c_i'$. We view $c_i'$ as the positive pair of the image $v_i$ in $\mathbb{R}^k$ space. We first project $z_{v_i}$ to the same dimensional space through another MLP layer $f_3 : \mathbb{R}^d \to \mathbb{R}^k$, $z_{v_i}' = f_3(z_{v_i}; \phi)$, and then calculate the contrastive loss:

$$\mathcal{L}_{score} = -\log\left(\sum_{i=1}^{n} \frac{e^{\text{sim}(z_{v_i}', c_i')/\tau}}{\sum_{j=1, j\neq i}^{n} e^{\text{sim}(z_{v_i}', c_i')/\tau}}\right). \tag{5}$$

$\mathcal{L}_{emb}$ and $\mathcal{L}_{score}$ execute contrastive learning at different layers, which controls the quality of graph structures in various aspects. More specifically, $\mathcal{L}_{emb}$ controls the alignment and uniformity between image embeddings and concept graph embeddings, and $\mathcal{L}_{score}$ makes concept vectors better hidden representations of input images Wang & Isola (2020). We combine the two contrastive learning objectives as well as a regularization term to control the sparsity of the learned graph and obtain the final contrastive learning loss

$$\mathcal{L}_{contrast} = \mathcal{L}_{emb} + \mathcal{L}_{score} + \beta L\text{-}1(\mathcal{A}), \tag{6}$$

where $L$-1 stands for the L-1 norm regularization and $\beta$ is the hyperparameter. Optimizing this loss gives us the latent graph $\mathcal{A}$ and the GNN parameters, we can then derive the updated concept score $c_i'$.

### 3.2.2 LABEL-FREE & CONCEPT-SUPERVISED OBJECTIVES

When datasets lack concept annotations for each instance, we would refer to them as *label-free* settings. Once we have the updated concept scores $c_i'$, we can use them to build the Graph CBM. There are different ways to build the layer from concepts to labels based on different backbones. One way is to use the new concept scores to replace the concept matrix in LF-CBM. Then we follow the design of LF-CBM and train a backbone model (RN50) with a transformation layer to get the concept bottleneck layer and the final sparse prediction layer. We call this model Graph LF-CBM.

In addition to separating the contrastive learning phase and the concept-based classification phase, we can also combine the losses and train them together:

$$\mathcal{L} = \text{CE}(\hat{y}_i, y_i) + \alpha(\mathcal{L}_{emb} + \mathcal{L}_{score}) + \beta L\text{-}1(\mathcal{A}), \tag{7}$$

$\alpha$ and $\beta$ are two hyperparameters adjusting the impacts of regularization. Instead of using the complex design of LF-CBM, here we build a linear classification layer from $c_i'$ to the label: $y_i = f_2(c_i')$. It can be seen as a variant of PCBM with concept graphs, so we call it Graph PCBM.

When there are concept supervisions, i.e., *concept-supervised* settings, the model learns the concept information purely according to the ground truth annotations. In this case, we do not need a language encoder to extract concept embeddings, so the $\mathcal{L}_{contrast}$ only depends on $\mathcal{L}_{score}$ (equation 4) and $L$-1$(\mathcal{A})$ (L-1 normalization controls latent graph complexity). We express the final loss function for models with concept annotations below:

$$\mathcal{L} = \text{CE}(\hat{y}_i, y_i) + \text{BCE}(\hat{c}_i, c_i') + \alpha\mathcal{L}_{score} + \beta L\text{-}1(\mathcal{A}), \tag{8}$$

where the $\text{BCE}(\cdot, \cdot)$ is the binary cross-entropy loss. Since concept annotations capture the concept relations or interactions, the BCE term will impact the latent graph to realize the ground truth relations to some extent, and the $\mathcal{L}_{score}$ also regularizes the latent graph by viewing the $c_i'$ as a hidden representation of image $v_i$. Combining them, the model can learn a good quality latent graph structure.

| Base experiments | CUB | Flower102 | HAM10000 | Cifar-10 | Cifar100 |
|---|---|---|---|---|---|
| LF-CBM | 73.90% (±0.28%) | 84.77% (±0.59%) | 66.76% (±0.43%) | 86.40% (±0.10%) | 65.16% (±0.14%) |
| Graph-(LF-CBM) | **75.59%** (±0.18%) | **86.00%** (±0.98%) | **67.47%** (±0.61%) | **86.54%** (±0.15%) | **65.96%** (±0.16%) |
| PCBM | 73.84% (±0.16%) | 79.01% (±1.19%) | 77.61% (±0.60%) | 95.71% (±0.07%) | 80.02% (±0.39%) |
| Graph-PCBM | **77.14%** (±0.40%) | **89.25%** (±0.69%) | **78.50%** (±0.52%) | **95.95%** (±0.09%) | **80.86%** (±0.26%) |

Table 1: **Graph CBM can better capture image information.** We report the mean and standard deviation from 10 runs with different random seeds.

## 4 EXPERIMENTS & RESULTS

### 4.1 SETUP

**Datasets**: For evaluating Graph CBMs, we choose various real-world datasets ranging from common objects to dermoscopic images. As the number of concept-supervised datasets is limited, We adopt label-free settings Oikarinen et al. (2023) and PCBM Yuksekgonul et al. (2023) for other general image classification tasks, in which we use multi-modal encoders to generate concepts for each downstream dataset. We introduce 1) common objects: CUB Wah et al. (2011), Flower102 Nilsback & Zisserman (2008), CIFAR-10 & CIFAR-100 Krizhevsky et al. (2009), and AwA2 Xian et al. (2018); and 2) medicial domains: HAM10000 Tschandl (2018) and ChestXpert Irvin et al. (2019). Dataset details can be found in appendix A.1.

Since many datasets do not contain concept annotations, we will use LLMs to generate and filter the concept set for each dataset. In CUB and CIFAR datasets, we use the same concept sets provided by Oikarinen et al. (2023). Furthermore, we found that concepts can be redundant and noisy in the CIFAR datasets. To mitigate these effects, we apply the submodular filtering algorithm proposed by Yang et al. (2023) to shrink the concept set to 30 concepts for CIFAR-10. For Flower and HAM10000, we use the selected concepts in Yang et al. (2023) and filter them with LLMs to reduce the number of concepts.

**Baseline**: We choose the state-of-the-art CBM models that do not need concept annotations during training, i.e., LF-CBM[2] Oikarinen et al. (2023) and Post-hoc CBMs (PCBM) Yuksekgonul et al. (2023). For multi-modality encoders, we choose the standard CLIP Radford et al. (2021). For Label-free CBMs, we use CLIP(ViT-B/16) image encoder as the backbone for FLower, HAM10000, CIFAR, and CUB. The numbers of concepts for datasets are as follows: 370 for CUB, 108 for Flower, 48 for HAM10000, 30 for CIFAR-10, and 50 for CIFAR-100. For CIFAR-10 and CIFAR-100 with the PCBM backbone, we perform submodular filtering to reduce the number of concepts from 143 and 892 to 30 and 50, while LF-CBM experiments will keep 143 and 892 concepts intact. In the concept-supervised setting, standard CBMs Koh et al. (2020) and CEMs Espinosa Zarlenga et al. (2022) serve as our baseline, and we still use CLIP(ViT-B/16) to extract features on CUB (112 concepts) and AwA2 (85 concepts) dataset. For ChestXpert (11 concepts), we use BioViL Bannur et al. (2023) as the image encoder, as BioViL is pre-trained on large X-ray datasets. We use the Adam optimizer and cosine scheduler during the training process.

### 4.2 RESULTS UNDER LABEL-FREE SETTINGS

**In vanilla target prediction, Graph CBM can outperform the state-of-the-art CBMs.** Evaluating test task accuracy reveals that Graph PCBMs are more accurate across tasks. On the other hand, Graph LF-CBMs learn, through GNN message passing and self-supervised contrastive learning, a better concept matrix that serves as a more expressive concept score initialization for third-party encoders. Our results suggest that including a latent graph gains higher task performance even with concept incompleteness – a highly desired property for real-world deployment. Finally, the performance improvements also validate our assumption that concept combination can act as a data

---

[2]we used CLIP(RN50) image encoder as the backbone for CIFAR, Flower102, HAM10000, and ResNet-18 trained on CUB from imgclsmob for CUB.

| Method | CUB | | AwA2 | | ChestXpert | |
| --- | --- | --- | --- | --- | --- | --- |
| | Label | Concept | Label | Concept | Label | Concept |
| CBM | 78.45% | **70.40%** | 95.24% | **97.48%** | 66.40% | **83.41%** |
| Graph-CBM | **80.03%** | 68.33% | **95.34%** | 97.05% | **66.82%** | 83.20% |
| CEM | 80.86% | 61.34% | 95.21% | 98.16% | 66.73% | 77.93% |
| Graph-CEM | **81.11%** | **61.53%** | **95.49%** | **98.62%** | **66.93%** | **78.27%** |

Table 2: Comparison between label prediction and concept prediction. We report the average accuracy (for label prediction) and roc-auc (for concept prediction) from 10 random seed experiments.

augmentation technique. In section A.4, we show more results on larger-scale datasets and the latent graph is still effective in enhancing prediction and intervention. We provide a detailed analysis of the importance of contrastive terms in appendix A.11.

### 4.3 RESULTS UNDER CONCEPT-SUPERVISED SETTINGS

**Graph CBMs can generalize their enhancement to concept-supervised settings.** In table 2, adding the latent graph can increase the label prediction performance for all the datasets from different domains, and the improvement is also universal towards different model architectures. Besides assisting in making better label predictions, including latent graphs can also match or even surpass the baseline's capability on concept prediction. It is important to note that the latent graph is primarily designed to leverage interactions between concepts to improve label prediction and enable more effective interventions, rather than directly optimize concept-level predictions. Nevertheless, our models still demonstrate notable improvements in some cases—particularly, Graph-CEM (G-CEM) not only boosts label prediction accuracy over standard CEM but also narrows the gap in concept prediction performance compared to CBM-based approaches.

### 4.4 COMPARISON WITH SOTAs

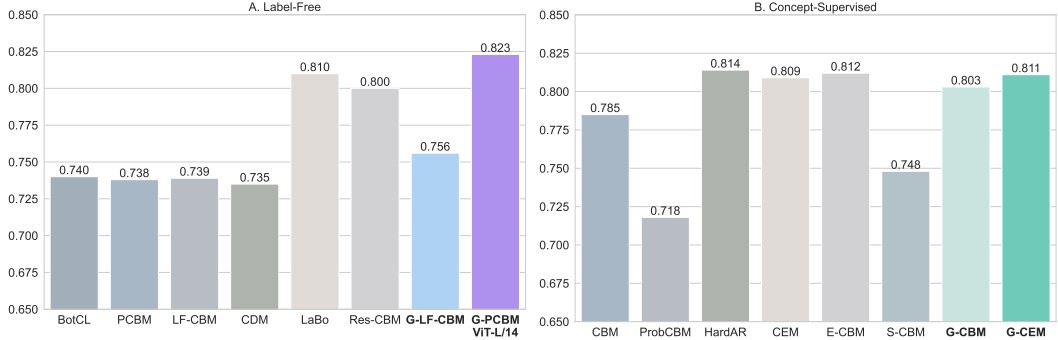

Figure 2: We compare our models with current SOTA results for corresponding training and dataset settings, i.e., label-free and concept-supervised. We report label accuracy in both subfigures.

In figure 2, we compare our proposed method with current state-of-the-art models and show the effectiveness of constructing a latent concept graph. We include a few more baseline models: BotCL Wang et al. (2023), CDM Panousis et al. (2023), LaBo Yang et al. (2023), and Res-CBM Shang et al. (2024) for label-free setting; ProbCBM Kim et al. (2023), HardAR Havasi et al. (2022), E-CBM Xu et al. (2024), and S-CBM Vandenhirtz et al. (2024) serves for concept-supervision setting. Because different baseline models are tested on various datasets, we select the CUB dataset as the common benchmark to present the comparison. CDM and S-CBM are reproduced using their code, and other baseline results are cited from their papers.

Under the label-free setting, LaBo Yang et al. (2023) and Res-CBM Shang et al. (2024) obtain the best two results among the chosen baselines; however, LaBo and Res-CBM focus on setting up an

advanced concept candidate bank, and they typically utilize much more concepts: for example, LaBo selects 50 concepts for each label class and results in total 10000 concepts for the CUB dataset. We make a fair comparison by using the same backbone image encoder here, and we select 1 concept per label class to form the final training and testing dataset. As shown in figure 2 A, utilizing a stronger backbone model can substantially enhance our G-PCBM performance ($77.1\% \rightarrow 82.3\%$) with fewer concepts (200 concepts in total) and training epochs (50 epochs; keep other settings the same). This indeed proves the effectiveness and efficiency of using latent graphs inside CBMs.

When the dataset contains the ground truth concept annotations, G-CBM and G-CEM can match up with the current SOTA results. HardAR Havasi et al. (2022) and E-CBM Xu et al. (2024) both work on uncovering hidden correlations among concepts by taking different approaches. HardAR focuses on generating good concept scores beforehand, and our latent graphs can plugin into their frame and continue looking for meaningful concept relations through message passing. Conceptually, the autoregressive method in HardAR admits the sequential dependency property, while G-CBM and G-CEM are concept-permutation invariant and more flexible to capture complex correlations. On the other hand, there are significant differences between our method and E-CBM. Rather than implicitly modeling concept correlation through joint energies or conditional probabilities, the latent graph provides us with explicit graph structures, which makes it easier to visualize and interpret the learned representations. The latent graph not only helps to learn a better model at training time but also makes the intervention more effective (see section 4.5).

The method of using latent graphs is independent of model architecture and orthogonal to baseline approaches. As shown by performance enhancements in tables 1 and 2, learning latent graphs offers a robust generalization ability across different model architectures and training settings, which further supports the expectation of latent graph effectiveness on other SOTAs (in A.7, we provide a case study on CDM to further support this statement). More importantly, unlike Havasi et al. (2022); Xu et al. (2024); Vandenhirtz et al. (2024) that capture concept correlations only on datasets with concept annotations, the latent graph is more **versatile**, as one can apply it in any CBM backbone in different training settings (label-free or concept supervised).

## 4.5 INTERVENTION DYNAMICS

The most important property of CBMs is the accessibility for users to adjust the concept activation to correct the model's false prediction and make the model more trustworthy. Therefore, besides improving label prediction right after training, we are also interested in the effectiveness of latent graphs when there are concept interventions. We follow the UCP policy Shin et al. (2023) to select concepts and design a simple method called *Lazy Intervention* A.5 to adjust the activation value.

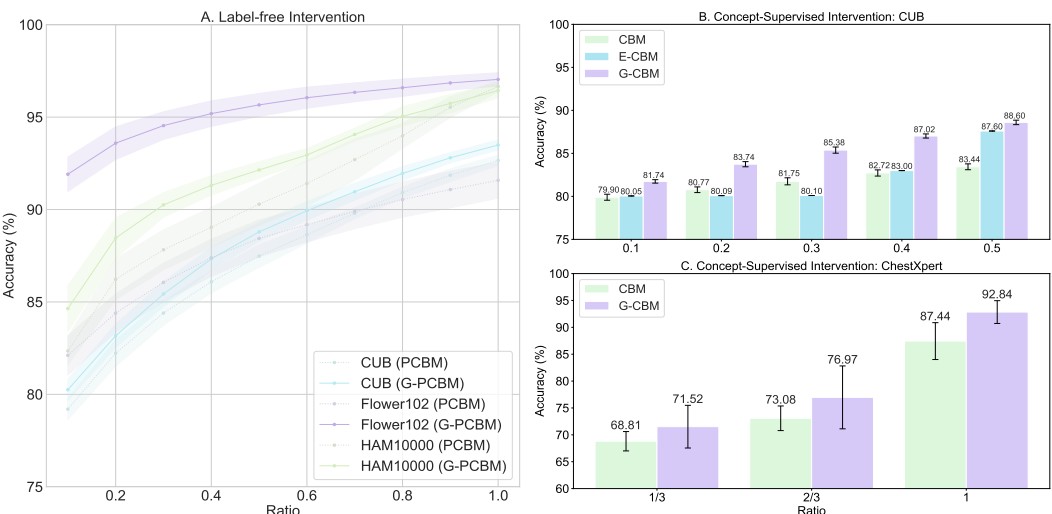

Figure 3: This figure shows the effectiveness of using latent graphs when intervening concepts. Subfigure A compares the after-intervention performance between PCBM and G-PCBM across three datasets under the label-free setting; subfigures B and C select concept-supervised datasets, and we compare G-CBM with CBM at various intervention ratios.

**Both label-free and concept-supervised settings, involving a latent graph will benefit the model in terms of label prediction when one intervenes or corrects the concept activation.** As shown in equation 1, we have two concept activation vectors $c$ and $c'$. The intervention takes place on $c$, so the new add-on values can also appreciate the latent graph structure to influence their neighborhoods to make a more prolonged impact. In this case, even though we fix the number of intervened concepts for baselines and our proposed methods, models with latent graphs intervene more concepts implicitly through message passing, and figure 3 also validates such message passing is positive. For the label-free setting, we select CUB and HAM10000 which are the dataset covering a general object dataset (birds) with more concepts (370 concepts) and the dataset from a medical domain with relatively fewer concepts (48 concepts). As shown in figure 3 A, G-PCBM can outperform or at least match up with the baseline for all intervention ratios.

Under the concept-supervised setting (figure 3 B & C), we show the comparisons between the CUB dataset and the ChestXpert dataset, where the ChestXpert dataset is a much smaller dataset regarding the number of concepts (11 concepts). We choose to intervene in a small number of concepts as it is a typical scenario. Unlike E-CBMs[3], whose intervention requires multiple energy calculations and gradient backpropagations, G-CBMs can perform one-step intervention and reach high after-intervention accuracy. We postulate that when concepts are supervised, latent graphs are also implicitly supervised by the ground truth concept relation; therefore, when intervening with concepts, the model gains more information from the ground truth concept relation and can enhance the label prediction more coherently. Moreover, the effectiveness of including a latent graph when intervening concepts further validates that the correlation information captured by the latent graph is positively meaningful. Full intervention comparisons are in appendix A.9.

### 4.6 How to Interpret The Concept Graph

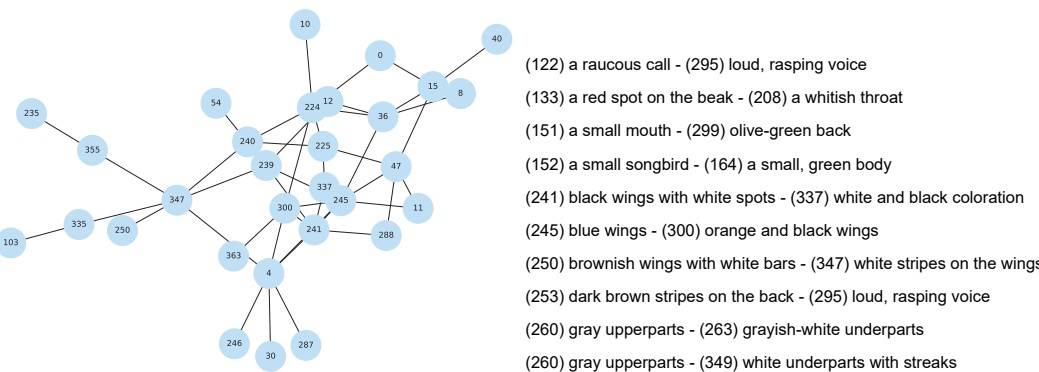

(122) a raucous call - (295) loud, rasping voice

(133) a red spot on the beak - (208) a whitish throat

(151) a small mouth - (299) olive-green back

(152) a small songbird - (164) a small, green body

(241) black wings with white spots - (337) white and black coloration

(245) blue wings - (300) orange and black wings

(250) brownish wings with white bars - (347) white stripes on the wings

(253) dark brown stripes on the back - (295) loud, rasping voice

(260) gray upperparts - (263) grayish-white underparts

(260) gray upperparts - (349) white underparts with streaks

Figure 4: We select a subgraph from the original concept latent graph of Label-free trained CUB, and we list a set of connections in the right to indicate the latent graph can uncover meaningful correlations among concepts. In Figure 13, we present the overall connectivity and concept correlations.

Figure 4 shows part of the latent graph under label-free settings for the CUB data and some examples of connected concept pairs. To further justify the validity and effect of the learned graph, we compare it with some "ground-truth" graphs.

**Our learnable graph can perform as well as the ground true concept graph generated from LLMs.** We ask ChatGPT Ouyang et al. (2022) to generate the most correlated concept pairs (in practice, we generate 50 concept pairs for the CUB dataset) as the real-world concept graph. When replacing the learnable graph with the ground true concept graph, Graph-PCBM can reach **77.69%** on the CUB dataset. Graph-PCBM with learnable graphs achieves **77.14%** accuracy, which is comparable to the ground true concept graph result. Moreover, learnable graphs will capture different information from LLMs: LLMs are good at connected concepts describing similar features based on body parts and color (like 'a black and white body' - 'a black and white color scheme'), while learnable graphs can recognize correlations other than visual similarity (like 'a loud, harsh cry' - 'a raucous call').

---

[3]Numbers for ECBM in figure 3 B are selected from Xu et al. (2024).

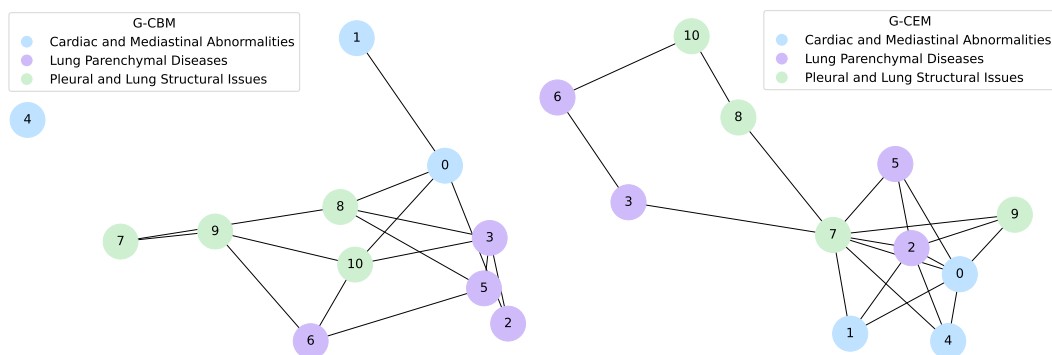

Figure 5: In the ChestXpert dataset, G-CBM and G-CEM can capture similar interactions among concepts given heterogeneous model architectures.

**Latent graph structure offers better robustness under concept attacks**. In figure 7, we empirically illustrate that concepts enclosed inside the latent graph are more robust under different attacks, while isolated concepts are more fragile. Such a phenomenon assumes that latent graphs can help recover the masked or perturbed concepts by aggregating their neighborhood information, indicating the benefits of latent graphs as well. Please view appendix A.6 for more detailed setup and analysis.

**When the dataset contains concept annotations, our latent graph discovers the true correlations among concepts and finds other relations.** As shown in figure 5, node 6 (*Pneumonia*) and node 10 (*Pleural Other*) are connected both in G-CBM and G-CEM. We extract a ChatGPT's answer by asking the relation between *Pneumonia* and *Pleural Other*:

> Pneumonia can directly affect the pleura, leading to conditions like pleural effusion, empyema, or pleurisy. These pleural complications often arise due to inflammation or infection spreading from the lungs to the pleural space, making the relationship between pneumonia and pleural disease significant in both diagnosis and treatment.

Therefore, the latent graphs learned under the label-free and concept-supervised settings both admit meaningful and reasonable connectivity among nodes, yielding better interpretability for models. In appendix A.8, we can further interpret the latent graph by extracting the salient subgraph.

Besides latent graphs finding meaningful correlations, different training settings result in heterogeneous graph expressions. As shown in figure 12, the label-free setting tends to learn a sparse graph while the concept-supervised setting prefers a dense graph. We postulate the difference might be attributed to the way concepts are collected. We provide a simple analysis in appendix A.10.

## 5 LIMITATIONS

One limitation of our Graph CBMs is that we only learn the latent interactions of the concepts. However, the learned graphs cannot show the hierarchical structure or more complex relationships among concepts. We believe utilizing external knowledge about concept relationships should be useful in building more interpretable concept graphs. We will also extend the work and explore the hierarchies of concepts instead of simply interacting with them.

## 6 CONCLUSIONS

In this paper, we have presented Graph CBMs: a simple yet effective method to integrate graph structures for capturing intrinsic correlations among concepts. Graph CBMs are orthogonal to previous CBM methods and can be plugged into those models to facilitate task performances while retaining interpretability. Graph-specific intervention policies also assist us to better sample concepts that should be intervened. In addition, our learnable graph performance can match up with that of a ground true concept graph. Future work may include 1) combining external knowledge graphs to assume the graph distribution and do the graph sampling, and 2) applying latent graph ideas to develop explanations for individual neurons inside CBMs Oikarinen & Weng (2024).

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

## A APPENDIX

### A.1 DATASET INTRODUCTION

- Caltech-UCSD Birds-200-2011, CUB Wah et al. (2011): CUB is the most widely-used dataset for fine-grained visual categorization tasks. It contains 11,788 images of 200 subcategories belonging to birds, we follow the same data processing as done in the Label-free-CBM Oikarinen et al. (2023) setting to select 5,990 images for the training set and 5,790 images for the validation set. For the concept supervised setting, we process the same way as Koh et al. (2020) and Espinosa Zarlenga et al. (2022), selecting 112 attributes as the concepts and use the same data splits.

- Oxford 102 Flower Nilsback & Zisserman (2008): is an image classification dataset comprising 102 flower categories. The flowers chosen to be flowers commonly occur in the United Kingdom. Each class consists of between 40 and 258 images. We follow the torchvision dataset setting maintainers & contributors (2016) and only use the training set for training and the validation set for testing.

- CIFAR-10 & CIFAR-100 Krizhevsky et al. (2009): The CIFAR-10/100 dataset (Canadian Institute for Advanced Research, 10 classes) is a subset of the Tiny Images dataset and consists of 60,000 32x32 color images. CIFAR-10 labels images with one of 10 mutually exclusive classes, and CIFAR-100 has 100 classes grouped into 20 superclasses. There are 6000 images per class with 5000 training and 1000 testing images per class in CIFAR-10, while CIFAR-100 divides the dataset into 500 training images and 100 testing images per class.

- HAM10000 Tschandl (2018): A dataset of 10000 training images for detecting pigmented skin lesions. It contains 7 labels and a representative collection of all important diagnostic categories in pigmented lesions HAM10000 provides 10,000 training images and 1,511 testing images.

- AwA2 Xian et al. (2018) is a zero-shot learning dataset containing 37,322 images and 50 animal classes. Unlike the CUB dataset Wah et al. (2011) in which concepts are defined at the instance level, images under the same label inside AwA2 Xian et al. (2018) will share the same concepts. We use all 85 attributes as concepts.

- ChestXpert Irvin et al. (2019) is a large dataset of chest X-rays and competition for automated chest X-ray interpretation, consisting of 224,316 chest radiographs of 65,240 patients, which features 14 uncertainty observations and radiologist-labeled reference standard evaluation sets. ChestXpert Irvin et al. (2019) does not provide binary label classification, so we cluster 11 out of 14 observations into 3 categories. The detailed data processing can be found in appendix A.3.

## A.2 CONFIGURATION AND RUNNING ENVIRONMENTS

| dataset | training epochs | Learning Rate | $\alpha$ | $\beta$ |
|---------|-----------------|---------------|----------|---------|
| CUB | 100 | 1e-3 | 0.1 | 0.2 |
| Flower102 | 500 | 1e-3 | 0.1 | 0.05 |
| HAM10000 | 100 | 1e-3 | 0.1 | 0.05 |
| CIFAR-10 | 100 | 1e-3 | 0.1 | 0.1 |
| CIFAR-100 | 100 | 1e-3 | 0.1 | 0.05 |

Table 3: Training Configuration for Graph LF-CBM

| dataset | training epochs | Learning Rate | $\alpha$ | $\beta$ |
|---------|-----------------|---------------|----------|---------|
| CUB | 100 | 1e-3 | 0.1 | 0.2 |
| Flower102 | 100 | 1e-3 | 0.1 | 0.01 |
| HAM10000 | 100 | 1e-3 | 0.1 | 0.05 |
| CIFAR-10 | 50 | 1e-3 | 0.1 | 0.05 |
| CIFAR-100 | 30 | 1e-3 | 0.1 | 0.05 |

Table 4: Training Configuration for Graph PCBM

| dataset | training epochs | Learning Rate | $\alpha$ | $\beta$ |
|---------|-----------------|---------------|----------|---------|
| CUB | 100 | 1e-3 | 0.05 | 0.0 |
| AwA2 | 50 | 1e-3 | 0.01 | 0.0 |
| ChestXpert | 50 | 1e-3 | 0.03 | 0.0 |

Table 5: Training Configuration for Graph CBM

| dataset | training epochs | Learning Rate | $\alpha$ | $\beta$ |
|---------|-----------------|---------------|----------|---------|
| CUB | 100 | 1e-3 | 0.07 | 0.0 |
| AwA2 | 50 | 1e-3 | 0.07 | 0.0 |
| ChestXpert | 50 | 1e-3 | 0.07 | 0.0 |

Table 6: Training Configuration for Graph CEM

We run all the experiments on a single GPU (NVIDIA A100). The GPU memory for Graph LF-CBMs and Graph PCBMs is less than 10GB with a batch size of 512 for all datasets. The full training run takes from 3 minutes to 2.5 hours depending on the dataset size and the number of training epochs. In practice, Graph LF-CBMs trained on CIFAR-100 take 2.5 hours, while Graph PCBMs trained on Flower need less than 3 minutes for execution. Graph CBMs and Graph CEMs on average take 8∼12 mins to finish training on the CUB and ChestXpert datasets, while they spend about 3∼5 mins on the AWA2 dataset. We also offer time measurements in Table 7: since the latent graph introduces more computational units, the model will be unavoidable to spend more time on training.

| Model | Training | Inference |
|--------|----------|-----------|
| PCBM | 5.7it/s | 2.09it/s |
| G-PCBM | 9.43it/s | 2.12it/s |
| CBM | 6.12it/s | 2.90it/s |
| G-CBM | 7.89it/s | 2.57it/s |

Table 7: Time Measurement between our proposals and baselines

### A.3 DATA PROCESSING FOR CHESTXPERT

CheXpert Irvin et al. (2019) uses "No Finding" to indicate the abnormality of patients' chest radiographs, and it is highly unbalanced. We will then cluster CheXpert's concepts into 3 different labels, and we will use the frontal and lateral X-ray images for each patient. The concepts and labels are classified in this way:

- **Group 1: Cardiac and Mediastinal Abnormalities**:
    - Enlarged Cardiomediastinum
    - Cardiomegaly
    - Edema (related to heart conditions)
- **Group 2: Lung Parenchymal Diseases**:
    - Lung Opacity
    - Lung Lesion
    - Consolidation
    - Pneumonia
- **Group 3: Pleural and Lung Structural Issues**:
    - Atelectasis (collapse of lung tissue)
    - Pneumothorax (air in pleural space)
    - Pleural Effusion
    - Pleural Other

If one patient meets multiple abnormal conditions (multi-labeled), we will select the most significant abnormality. We choose not to include the *Fracture* observation, as it forms a cluster itself and shares no commonalities with other observations.

### A.4 PERFORMANCE ON OTHER LARGER-SCALE DATASETS

| Method | Places365 | | | ImageNet | | |
|--------|-----------|----------|---------|----------|----------|---------|
| | Acc | $\alpha$ | $\beta$ | Acc | $\alpha$ | $\beta$ |
| PCBM | 55.24% ($\pm0.08\%$) | - | - | 77.49% ($\pm0.10\%$) | - | - |
| Graph-PCBM | **55.31%** ($\pm0.08\%$) | 0.1 | 0.15 | **78.48%** ($\pm0.10\%$) | 0.1 | 0.17 |

Table 8: We report the average accuracies from 3 different random seed experiments along with the standard deviation.

In this section, we investigate the performance enhancement in prediction and intervention brought by latent graphs on large-scale datasets. We choose *Places365* López-Cifuentes et al. (2020), a scene recognition dataset composed of 10 million images comprising 434 scene classes, and *ImageNet* Deng et al. (2009) which contains 14,197,122 annotated images according to the WordNet hierarchy. We train G-PCBMs on these two datasets with 10 epochs, 1024 batch size, learning rate at 0.01, and Adam optimizer.

In Table 8, the G-PCBM has a latent graph to help capture the intrinsic concept correlation and improve label prediction in both large-scale datasets. We continue to present the positive impacts of involving a concept latent graph inside the model on *ImageNet* as a case study for intervention. Figure

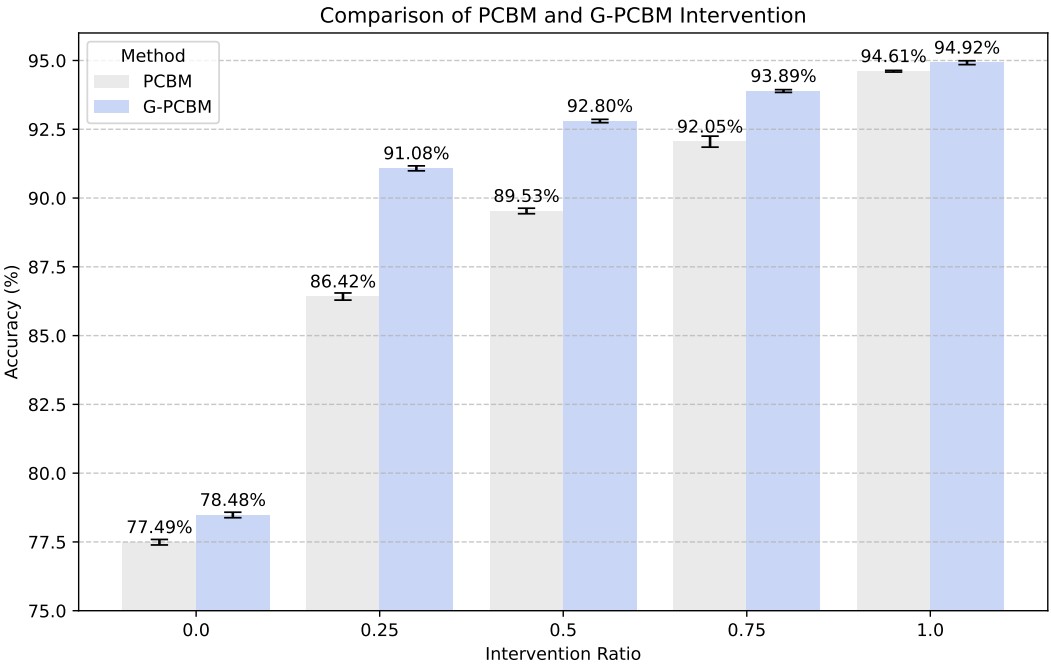

Figure 6: In a large-scale dataset, having latent graphs can still promote models to interact with intervention at different ratios.

6 also indicates the effectiveness of latent graphs in intervening large-scale datasets. In particular, G-PCBM significantly improves intervention performance in low intervention ratios. This also aligns with the results we conclude in the other datasets.

## A.5 LAZY INTERVENTION

**Lazy Intervention**: We define two sets of concept scores

$$\mathcal{R} = \{c_i \mid h(c_i') = y_i\}, \quad \mathcal{W} = \{c_i \mid h(c_i') \neq y_i\},$$

where $\mathcal{R}$ and $\mathcal{W}$ are sets of concept scores making right and wrong predictions. We can further partition them using class labels, so $\mathcal{R} = \cup_{i=1}^m \mathcal{R}^i$, and so does $\mathcal{W}$. We then define the difference set,

$$\mathcal{D} = \{\text{mean}(\mathcal{R}^j) - \text{mean}(\mathcal{W}^j) \mid 1 \leq j \leq m\},$$

$m$ is the number of classes. Each $d^j \in \mathcal{D}$ can be viewed as a prototype of the intervention vector for $j$-th class. The intervention procedure will be

$$\text{Intervention} = \{c_i + d^j \mid \forall c_i \in \mathcal{W}^j, 1 \leq j \leq m\}.$$

When the dataset contains concept annotations, the $\mathcal{R}$ records positively classified concepts, and the $\mathcal{W}$ records falsely classified concepts. At the intervention step, *Lazy intervention* will only intervene on falsely classified concepts.

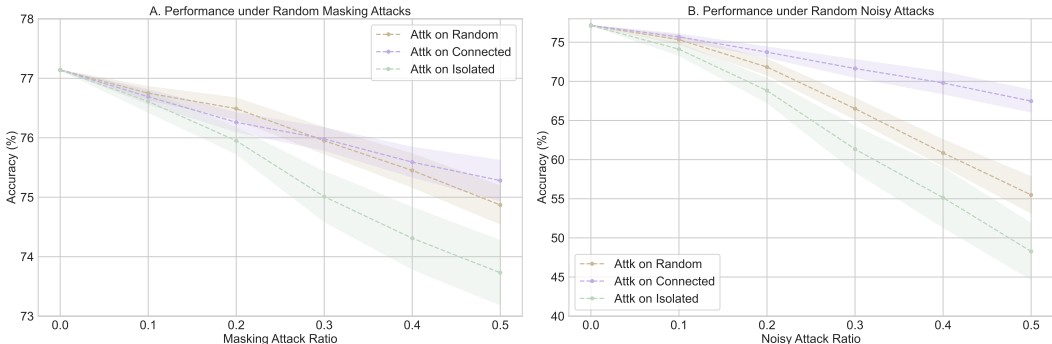

Figure 7: Attacking different types of concepts (nodes) in the CUB dataset can result in different scales of performance degradation.

### A.6 Latent Graphs Offer Robustness under Label-Free Settings

In this section, we further investigate the effect of latent graphs on concepts (nodes), and we design a simple experiment to test such impacts. We first partition concepts into connected concepts (node degree $\geq 1$) and isolated concepts, and we also keep the whole concept set as a baseline group for random attacks; then, we will randomly mask or perturb the same number of concepts in these groups separately; lastly, we let our model predict those corrupted concepts. We choose the CUB as the testing dataset for this experiment[4].

As shown in figure 7, attacking connected concepts does not harm the model performance as much as attacking isolated concepts or random concepts. We hypothesize that the latent graph structure provides much better robustness towards connected concepts, while the isolated ones cannot benefit from it. When masking or perturbing a concept, there will be information missing for the final prediction layer. Nevertheless, the latent graph structure can aggregate neighborhood information to recover the masked or perturbed concept so that the final prediction layer is still able to make a reasonable prediction.

### A.7 Latent Graphs Benefit Various CBM Backbones

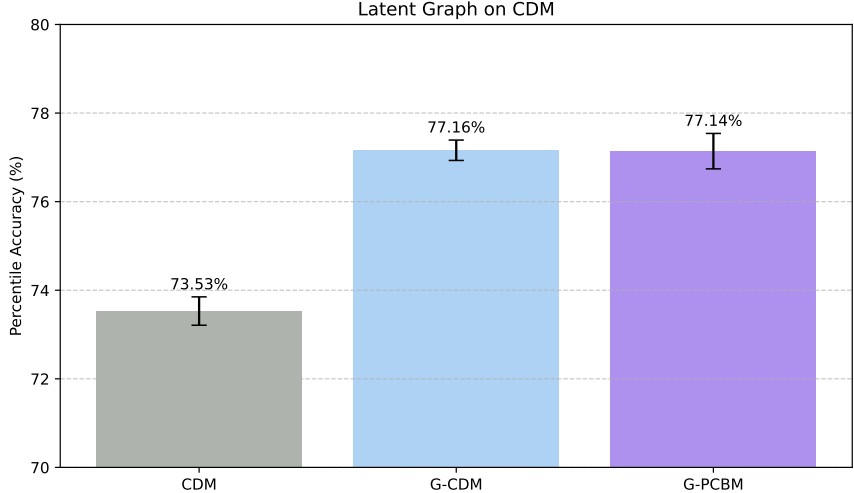

Figure 8: Adding latent graph to the CDM can significantly improve model performance.

In this section, we validate the effectiveness of latent graph on other CBM backbones. We choose CDM Panousis et al. (2023) as the case study to show the benefits of latent graphs. The most

---

[4]We select checkpoints with a similar amount of connected concepts and isolated concepts.

important unit in CDM Panousis et al. (2023) is the concept presence indicator which is modeled from a Bernoulli distribution, and the motivation behind the indicator variable is to sparsify the required concepts for label prediction. In Fig 8, applying latent graph to CDM can make a great enhancement in terms of label prediction, and latent graph can help to recover those filtered concept information to further boost model performance. The phenomenon is similar as we have discussed in A.6, as masking attack also sparsify concepts, and the latent graph can prevent the degradation caused by such attacks or operations. Along with the performance enhancements in Tables 1 and 2, learning latent graphs offers a robust generalization ability across different model architectures and training settings, which encourages the expectation of latent graphs effectiveness on other methodologies like LaBo, HardAR, and E-CBM.

## A.8    SALIENT SUBGRAPH FOR CONCEPT-SUPERVISED SETTINGS

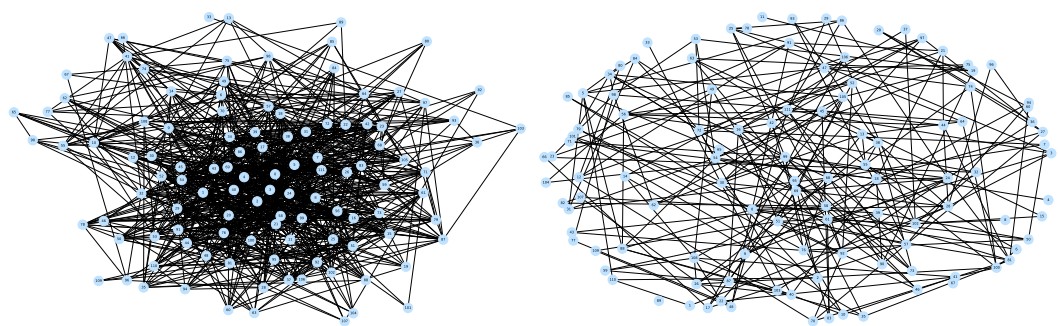

Figure 9: Both graphs are G-CBM concept graphs for the CUB dataset. Right: original concept graph. Left: salient subgraph structure.

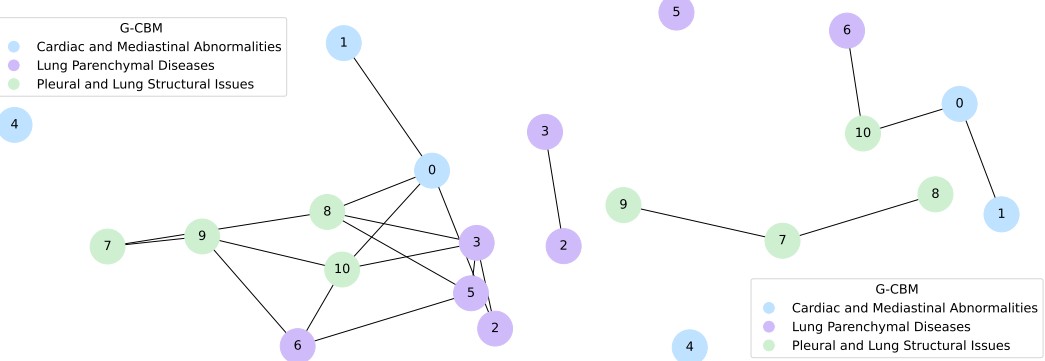

Figure 10: Both graphs are G-CBM concept graphs for the ChestXpert dataset. Right: original concept graph. Left: salient subgraph structure.

The concept-supervised setting favors dense graphs, but it also easily leads to redundant connectivity for the latent graph, which will harm the concept graph interpretability and inference efficiency. We follow a heuristic strategy to find the salient subgraph inside the original concept graph: we mask one edge at a time and check the performance; if the performance remains the same or goes up, we delete that edge from the original subgraph. By doing this, we can improve our label prediction marginally without sacrificing concept prediction (CUB label accuracy: 80.03% → **80.40%**; concept AUC: 83.20% → 83.20%; # of edges: 679 → **239**). Extracting the salient subgraph makes the concept graph more easily interpretable: for example in figure 10, we can filter out the redundant edges and draw a more sparse concept graph. In the salient subgraph, concepts belonging to the same label class are more likely to connect. This also demonstrates that our latent graph effectively captures the hidden concept correlation embedded in the concept supervision.

We again provide a quote from ChatGPT to show the reasonable connection between node 0 (Enlarged Cardiomediastinum) and node 10 (Pleural Other):

**Cardiac or vascular causes**: Enlarged cardiomediastinum often results from cardiac enlargement (e.g., heart failure or pericardial effusion) or vascular abnormalities (like aortic aneurysms). Some of these conditions can also cause pleural changes. For example:

1) Heart failure can lead to pleural effusion (fluid in the pleural space), which may manifest as a "Pleural Other" abnormality.

2) Aortic aneurysm or dissection may affect surrounding pleural structures due to proximity, causing pleural thickening or effusions.

**Malignancies**: Tumors in the mediastinum (e.g., lymphomas or metastatic disease) can enlarge the cardiomediastinum and simultaneously invade or affect the pleura, leading to pleural abnormalities.

**Infections and inflammatory conditions**: Severe infections like tuberculosis or mediastinitis can affect both the mediastinum and pleura, causing enlargement of the cardiomediastinum and pleural changes.

## A.9 G-CBM INTERVENTION AT FULL RATIO

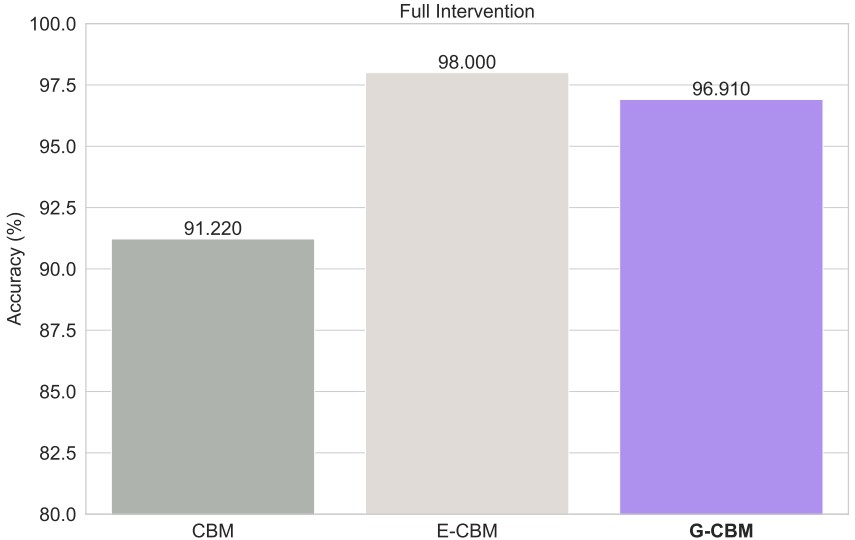

Figure 11: Comparison among models when full interventions.

We compare our G-CBMs with E-CBMs Xu et al. (2024) and standard CBMs Koh et al. (2020) at full interventions. In figure 11, G-CBMs can improve the performance significantly compared to the standard CBMs, and it can match up with E-CBMs as well. E-CBMs require energy calculations and gradient backpropagations to update the label probability, so the full intervention can offer sufficient information for E-CBMs to reach a good after-intervention performance. However, E-CBMs cannot handle low-ratio intervention well as shown in figure 3 B, while our G-CBMs can be effective at different intervention ratios coherently. Plus, one-step intervention makes G-CBMs more efficient.

## A.10 GRAPH COMPLEXITY

**Graph CBMs favor sparse graphs under the label-free settings.** Without concept supervision, Graph CBMs prefer sparse graphs in general. As shown in table 12, the model gains performance improvement as we continue making the learnable graph sparse. Dense graphs result in over-smoothing concept scores, while sparse graphs can better preserve each concept value and propagate concept relation reasonably. We compare different graph complexity effects on the CUB dataset by varying the hyperparameter $\beta$ (large $\beta$ indicates more sparsity in the graph) under the Graph PCBM setting. We observe that our Graph PCBMs favor sparse graphs. The reason might be due to the way one collects concepts. If the model is trained under the label-free setting, one relies on a sophisticated concept generator like LLMs; however, LLMs will easily provide lots of redundant and

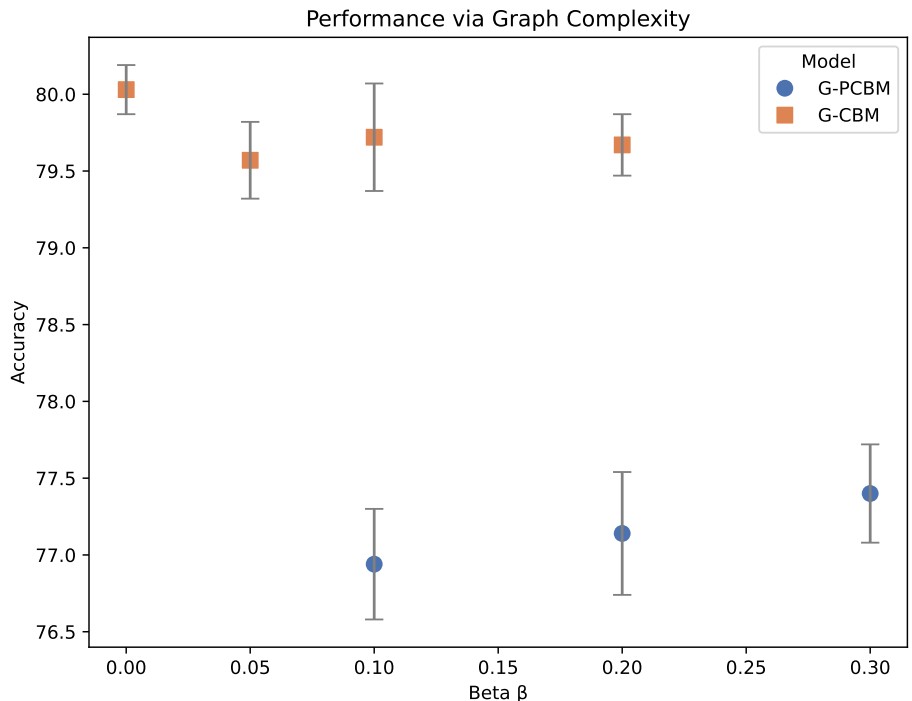

Figure 12: Performance changes as we set different $\beta$ value to control the latent graph complexity.

| Dataset | CUB | | | | HAM10000 | | | |
|---|---|---|---|---|---|---|---|---|
| $\mathcal{L}_{emb}$ | - | - | ✓ | ✓ | - | - | ✓ | ✓ |
| $\mathcal{L}_{score}$ | - | ✓ | - | ✓ | - | ✓ | - | ✓ |
| Performance | 73.90% | 75.38% | 74.11% | **75.59%** | 66.76% | 62.87% | 65.10% | **67.47%** |

| Dataset | Flower102 | | | | CIFAR100 | | | |
|---|---|---|---|---|---|---|---|---|
| $\mathcal{L}_{emb}$ | - | - | ✓ | ✓ | - | - | ✓ | ✓ |
| $\mathcal{L}_{score}$ | - | ✓ | - | ✓ | - | ✓ | - | ✓ |
| Performance | 84.77% | 76.78% | 85.99% | **86.00%** | 65.16% | 62.97% | 59.49% | **65.96%** |

Table 9: **Multi-level contrastive learning is crucial for self-supervised concept score qualities.** If we train Graph LF-CBMs with one excluding contrast loss, the model can fail to yield a good concept matrix; while considering both contrast losses, models can gain more benefits.

repeated concepts. On the other hand, if the dataset contains concept annotations, concept sets are usually smaller in terms of the number of concepts and sufficiently informative. Moreover, concept annotations can act as an implicit concept correlation supervision, and models are expected to capture those hidden correlations in nature.

## A.11 HOW CONSTRASTIVE TERMS AFFECT MODEL PERFORMANCE

In this section, we study how the contrastive learning loss impacts the results.

**Using both $\mathcal{L}_{emb}$ and $\mathcal{L}_{score}$ is crucial for Graph LF-CBM** In table 9, we examine the different combinations of contrast losses on CUB dataset. We observe that only taking $\mathcal{L}_{emb}$ increases the Graph LF-CBM marginally while using $\mathcal{L}_{score}$ can further improve model performance on the CUB dataset. However, solely using one of the contrast losses on the HAM10000 and CIFAR100 datasets fails to outperform the LF-CBM baseline. Last but not least, considering both contrast losses simultaneously yields the best accuracy for both tasks. If only $\mathcal{L}_{emb}$ is present, the learnable graph is extremely insensitive to the L-1 loss that controls graph complexities, resulting in a highly dense graph. On the other hand, just considering $\mathcal{L}_{score}$ will easily learn a super sparse graph. Therefore, having $\mathcal{L}_{emb}$ and $\mathcal{L}_{score}$ allows us to adjust the final graph complexity in practice better.

| Method (Graph PCBM) | CUB | HAM10000 | CIFAR-100 |
|---|---|---|---|
| w/o ($\mathcal{L}_{emb} + \mathcal{L}_{score}$) | 76.89% (±0.65%) | 77.66% (±0.41%) | 75.67% (±0.35%) |
| w ($\mathcal{L}_{emb} + \mathcal{L}_{score}$) | **77.14%** (±0.40%) | **78.50%** (±0.52%) | **80.86%** (±0.26%) |

Table 10: **Adding contrast loss to target supervision can help model performance.** Having the two contrast regularizers will lead to higher accuracy for label prediction.

**With target supervision, contrast regularizations further help the Graph PCBM gain expressivity.** The idea of contrastive learning is to provide us with a self-supervised training objective and express concepts as a different augmented view of images. It then questions us: will contrast regularization benefit Graph PCBM when there is a target supervision? We conducted experiments on CUB, HAM10000, and CIFAR-100, and the results are offered in table 10. Graph PCBM with $\mathcal{L}_{emb}$ and $\mathcal{L}_{score}$ perform well across all the datasets. Especially for CIFAR-100, Graph PCBM can achieve more than 5% enhancement. In Graph PCBM w/o contrast loss, the learnable graph fully relies on target supervision and L-1 regularizer, which makes it difficult to control graph complexity across tasks, as the graph becomes highly sensitive to the L-1 regularizer, and the target prediction loss may also influence graph variously.

## A.12 VISUALIZATION OF LEARNABLE GRAPH

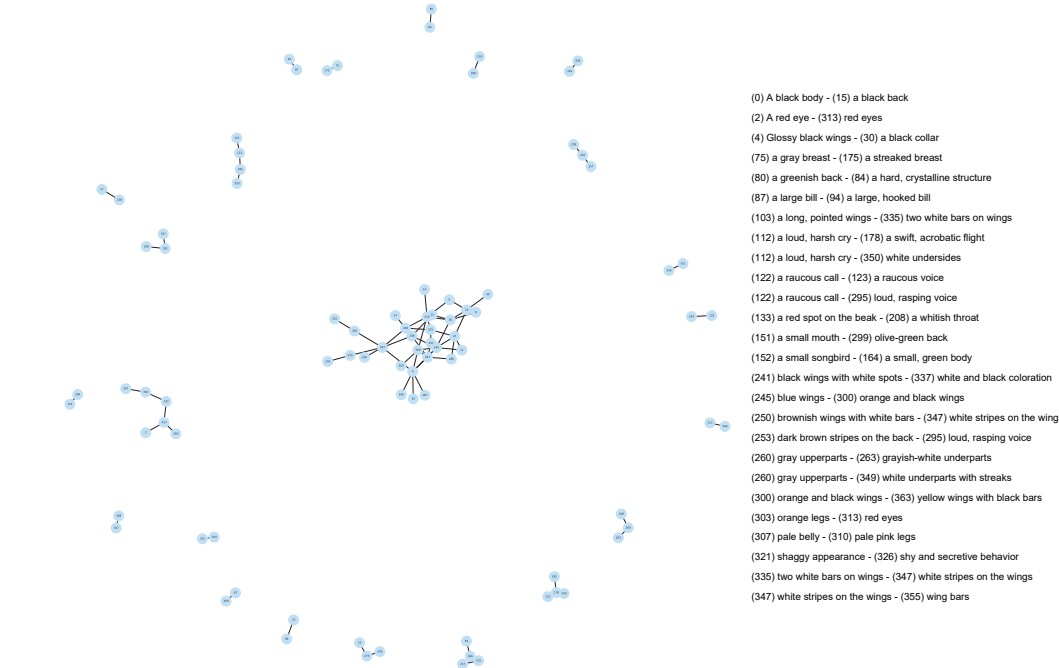

Figure 13: The overview of the CUB's concept graph in label-free settings.

