# OpenReview forum: "Graph Concept Bottleneck Models"
_ICLR.cc/2025/Conference — Submitted to ICLR 2025_

### Official Review · Reviewer_TaXb · 2024-10-28

**Soundness:** 3
**Presentation:** 3
**Contribution:** 3
**Rating:** 5
**Confidence:** 3

**Summary:**

This paper introduces a novel variant of the Concept Bottleneck Model (CBM), called Graph CBM, which aims to capture the intrinsic correlations between concepts by incorporating graph structures. By constructing an underlying concept graph, Graph CBM enhances interactions between concepts, thereby improving both the interpretability and performance of the model.

**Strengths:**

1. Innovation: The introduction of graph structures into the concept bottleneck model addresses the limitations of traditional CBMs that overlook concept relationships, significantly enhancing the model's interpretability and performance.

2. Broad Applicability: Graph CBM can be integrated with existing CBM methods, offering strong extensibility and improving performance across various CBM applications.

**Weaknesses:**

Although the use of graph structures helps explain relationships between concepts, it still falls short in providing interpretability for individual neurons within the model.

**Questions:**

See Weaknesses.

---

> ### Author Response · Authors · 2024-11-23
> **Response to TaXb**
>
> Dear Reviewer TaXb,
>
> Thanks for your valuable feedback and for acknowledging the innovation of this work and its broad applicability to integrate with existing CBM methods! Below, we would like to address your concerns and comments.
>
> Regarding the comment “*Although the use of graph structures helps explain relationships between concepts, it still falls short in providing interpretability for individual neurons within the model.*”, we believe that there might be some misunderstanding, please allow us to clarify below:
> - In this work, we focus on concept bottleneck models, whose goal is to build interpretable models by adding a concept bottleneck layer between the backbone feature and the predictor. The neurons in the concept bottleneck layer (CBL) are interpretable, i.e., each of the individual neurons in that layer corresponds to a concept, which is the basis of the final prediction.
> - Some work is focused on developing neuron interpretability tools [1,2,3] to identify the concepts of neurons, which has different goals than ours in this paper. In their setting, they are trying to identify the concepts of neurons given a model, and the model’s parameters are frozen. In our setting, we are training the model to make it intrinsically interpretable. Therefore, these are different problems.
> - In fact, their results are complementary to ours, i.e. we could use neuron-interpretability tools to further understand what’s happening in the backbone and concept bottleneck models. Moreover, in Label-free CBM, the authors actually use the neuron-interpretability tool (CLIP-Dissect [1]) in the training loop to learn the concept bottleneck layer.
>
> In short, we clarified that the CBM approach in this paper still has “individual neuron interpretability” in the Concept bottleneck layer, and one can use neuron interpretability tools to identify other layers in CBM to understand the neuron functionalities. We also clarified that the CBM approach and neuron interpretability tools are two **different but complementary** problems.
>
> To avoid confusion, we will add a paragraph in the revised draft discussing the connections between the CBM approach and neuron interpretability tools.
>
> Please let us know if you still have any questions or concerns, and we would be happy to address them!
>
> [1] Oikarinen, Tuomas, and Tsui-Wei Weng. "Clip-dissect: Automatic description of neuron representations in deep vision networks." arXiv preprint arXiv:2204.10965 (2022).
>
> [2] Bau, David, et al. "Network dissection: Quantifying interpretability of deep visual representations." Proceedings of the IEEE conference on computer vision and pattern recognition. 2017.
>
> [3] Hernandez, Evan, et al. "Natural language descriptions of deep visual features." International Conference on Learning Representations. 2021.

---

### Official Review · Reviewer_8TA6 · 2024-10-31

**Soundness:** 2
**Presentation:** 2
**Contribution:** 2
**Rating:** 6
**Confidence:** 2

**Summary:**

This paper presents Graph Concept Bottleneck Models (Graph CBMs), an advancement of Concept Bottleneck Models that incorporates learnable graph structures to capture interactions among concepts. Unlike traditional CBMs that assume concepts are independent, Graph CBMs model these relationships to enhance performance and maintain interpretability. Empirical results show that Graph CBMs outperform existing CBMs in image classification tasks, facilitate more effective interventions using concept graphs, and demonstrate robustness across various training settings and backbones.

**Strengths:**

1. The paper introduces Graph CBMs, which improve the interpretability of Concept Bottleneck Models by modeling interactions among concepts through a learnable graph structure.
2. The proposed contrastive learning framework can perform graph structure learning.
3. The proposed method is compatible with existing CBM approaches and improves performance across different backbones and training setups while maintaining interpretability.

**Weaknesses:**

1. The paper is challenging to follow because many design choices seem subjective with less clear, detailed explanations. I believe that the design processes of many components in the paper are not clearly explained, and the experiments do not sufficiently validate the claims. Further clarification is needed.
2. Figure 1 provides insufficient detail, as the graph calculation process is represented with a simplistic and unclear sketch, making it difficult to convey meaningful insights.
3. The rationale behind using the non-parameterized graph propagation mechanism is unclear. What is the actual meaning of the update rule of concept score?
4. The claim that the inverse matrix is hard to compute and may introduce unrelated long-range dependency noise requires further clarification.
5. Additionally, the meaning of the "one-step approximation" are not explained and there is no any citations.
6. Lines 175-178 contain an ambiguous statement: "As an intrinsic structure, we expect that the concept graph should not depend too much on the labels; otherwise, the graph cannot be learned well in label-incomplete settings." This sentence is unclear and very subjective. I can not see any reason why the contrastive learning method should be used next.
7. Lines 196-197, there's no definition of $W_l^{'}$.
8. The description of "However, $c_i$ only plays a marginal role in $\mathcal{L}_{emb}$ which limits the performance. ... we design a second contrastive loss" is hard to understand. What is the logic behind $c_i$ playing a marginal role and thus limiting performance?
9. The experimental results in Table 2 and Figure 2 seem marginal compared to traditional methods.
10. The design of the loss function (Equation 6) appears unnecessarily complicated, and the justification for this complexity is insufficient. The experiment in Appendix A.9, which examines how the two contrastive terms affect model performance, only includes two datasets. Using just two datasets to infer the necessity of joint training for these losses seems inadequate. Why weren't experiments conducted across all five datasets? The results of the complex loss design do not appear to be well validated, and the paper does not clearly demonstrate through insights or experiments why this design is necessary.

**Questions:**

Please refer to the Weaknesses.

---

> ### Author Response · Authors · 2024-11-23
> **Response to 8TA6 (1)**
>
> Dear reviewer 8TA6,
>
> Thank you for your comments and we appreciate the time and effort you put into it. We will try our best to address your concerns in this rebuttal.
>
> “*Lack of explanation.*”
>
> 1. We apologize that some of the context confuses you, and we will be grateful if you are willing to point out which parts cause the most confusion explicitly and clearly. For the design choice, the most important modules are the contrastive loss for latent graph structure learning and the message passing for final label prediction. For message passing, we assume it is clear and intuitive to use graphs to update the concept scores. As to contrastive learning, in lines 175-182, we explain detailedly why we decided to use contrastive losses. We briefly summarize that paragraph in the following bullet points.
>
> - i. Since the concept graph aims to capture intrinsic interactions among concepts, we hypothesize that the structure of the concept graph shall be label-agnostic. Thus, we adopt a self-supervised manner for concept graph structure learning, and contrastive learning has shown its supremacy in lots of self-supervised training.
> - ii. We assume that each image can be viewed as a combination of concepts; in other words, a combination of concepts should be an augmented view of an image. This will allow us to design positive pairs for contrastive learning.
> - iii. Concept score vectors are some hidden representations of input images.
>
> “*Figure 1 provides insufficient detail, as the graph calculation process is represented with a simplistic and unclear sketch, making it difficult to convey meaningful insights.*”
>
> 2. Figure 1 gives an overview of the intact training process, and we will make changes in the revision by replacing the graph with a more informative one. However, the training process is also presented formally in section 3 METHOD.
>
> “*The rationale behind using the non-parameterized graph propagation mechanism is unclear. What is the actual meaning of the update rule of concept score?*”
>
> 3. The non-parameterized propagation is only for message passing and score updating, and it is one of the common message passings in the GNN community and the mechanism for making graph impacts. In equation (1), we flatten it out, and one can easily find out that the non-parameterized propagation is simply a weighted skip connection. The skip connection prevents over-smoothing issues raised by aggregation and propagation in standard graph message passing.
>
> We would like to point out that the reviewer has made a hugely misdirected discussion. The goal and theme of our paper is to **discover concept correlation and make the CBM realize such correlation** to better make predictions and interact with intervention. Numerical efficiency behind message passing has **NEVER** been our focus, and we also offer many citations in our submission to those who might be interested (lines 162-163).
>
> “*The claim that the inverse matrix is hard to compute and may introduce unrelated long-range dependency noise requires further clarification.*”
>
> 4. Inverse matrix calculation involves matrix decompositions such as LU Decomposition and QR decomposition, and those decompositions will make the inverse calculation complexity $\mathcal{O}(n^{3})$. Even the most advanced methods like Coppersmith-Winograd [1] and Strassen-Based Inversion [2] will have complexity at roughly $\mathcal{O}(n^{2.376})$ and $\mathcal{O}(n^{2.81})$. The long-range dependency noise [3] comes from the calculation steps inside the inverse matrix algorithms, particularly because it refers to interactions or dependencies between distant elements, and noise refers to small perturbations or inaccuracies that can propagate and amplify through the inversion process.
>
> However, we would like to emphasize that this is **NOT** the main focus of our paper, and the concerns on numerical linear algebra are outside the scope of our study object. It is less relevant to what we desire to present and have presented in our submission. How to propagate the messages using the latent concept graph is only a model choice, and it can be changed in many different ways (take an analogy, you can use many different GNN models to learn the node representation in a graph and the choice of message passing mechanisms in the GNNs is not the focus in many downstream tasks).

---

> ### Author Response · Authors · 2024-11-23
> **Response to 8TA6 (2)**
>
> “*Additionally, the meaning of the "one-step approximation" are not explained and there is no any citations.*”
>
> 5. We define the one-step approximation in equation (1), and the reason for using a so-called one-step approximation is intentionally to keep the CBM framework computationally efficient and interpretable, aligning with the model’s purpose as a Concept Bottleneck Model rather than a graph-based optimization study. The purpose is to create a feasible approximation that retains the interpretability and expressiveness of the CBM without computational overhead that might obscure the model’s explanatory value. This is a standard approach in many GNN applications and is sufficient to meet our CBM objectives.
>
> In lines 162-163, we cite the paper [4] which explicitly deduces the one-step approximation or so-called APPNP, and our implementation basically follows their method.
>
> **Our primary focus is enhancing CBMs by introducing latent graph structures**, **NOT** on advancing matrix inversion techniques or exploring sophisticated graph-theoretic optimizations. The one-step approximation allows us to leverage graph structures for interpretability and model robustness in a way that is computationally tractable and aligns well with the CBM’s purpose.
>
>
> “*Lines 175-178 contain an ambiguous statement.*”
>
> 6. We apologize for the confusing writing and want to illustrate the rebuttal. Previous methods favor finding the mapping between label space and concept space through the prediction head, and they usually overlook the correlations inside concept space alone. The motivation and goal of our paper are to **discover the correlations only among concepts**, and then so-generated concept graphs can help the CBMs realize those correlations. Notice that we are not talking about the dependencies between labels and concepts. In this case, one should not use label information to construct the concept graph. Besides looking at the concept information inside the label space, the input image carries features and information about concepts as well. Therefore, a self-supervised manner for constructing the latent graph can effectively ensure the final graph quality while avoiding label-information leakage.
>
>
> In practice, the label-free training setting does not even provide concept supervision, which makes a contrastive learning mechanism essential in a label-free setting. By aligning graph representations with image embeddings, it learns a graph that generalizes well, even in settings where labels may be incomplete or noisy. This independence improves model robustness, a feature validated through experimentation.
>
> “*Lines 196-197, there's no definition of $W’_{l}$*”
>
> 7. We are sorry for the typo we made. It should be $W_{l}$. The $W_{l}$ is defined in lines 200-201: *$W_{l}$, is the learnable weight matrix.*
>
> “*What is the logic behind $c_{i}$ playing a marginal role and thus limiting performance?*”
>
> 8.
> - First, only containing $\mathcal{L}_{emb}$ gives a suboptimal performance as shown in Table 7 and Table 8.
> - Second, freezing the image and text encoders will hugely impact the performance and effectiveness of using $\mathcal{L}_{emb}$.
> - Third, the concept score $c_{i}$ only acts as a scale weight for each concept/text embedding, so the training objective will be more likely to adjust the embeddings rather than the concept score $c_{i}$.
>
> We agree that the writing here can be confusing, and we will rewrite the line 219-220 to: *However, only considering $\mathcal{L}_{emb}$ yields a suboptimal performance, and we have shown it empirically in table 7 and table 8.* And we will discuss the reason why having $\mathcal{L}_{emb}$ solely might be insufficient in appendix 9.
>
> “*The experimental results in Table 2 and Figure 2 seem marginal compared to traditional methods.*”
>
> 9. Evaluating the performance of CBMs involves many different aspects, and we still outperform the baseline in label prediction throughout the training settings and dataset benchmarks. One of the biggest advantages is the intervention capability. Section 4.5 presents that our proposed method can surpass the baseline and sota method significantly. This also supports our proposal's effectiveness.

---

> ### Author Response · Authors · 2024-11-23
> **Response to 8TA6 (3)**
>
> “*The design of the loss function (Equation 6) appears unnecessarily complicated, and the justification for this complexity is insufficient.*”
>
> 10. Regarding the "complex" training objective, we have indicated that those contrastive terms act on different granularity based on the assumptions:
>
> - i. The latent graph ought to be label-agnostic.
> - ii. Concept combination can be viewed as an augmentation of the input image
> - iii. The concept score vectors $c_{i}$ and $c'_{i}$ are also some hidden representations of the input image.
>
> Having those hypotheses, we can argue that
> - $\mathcal{L}_{emb}$ contrasts the graph and the image at the hidden model space level.
> - $\mathcal{L}_{score}$ contrasts the graph and the image at the concept space level.
>
> Empirical results in Appendix 9 have validated our postulates and effectiveness.
>
> Though the studies in Appendix 9 serve as ablation studies and case studies, we valued the reviewer’s opinion and suggestions. The completed ablation studies for Table 7 are below.
>
> |  Model |   Flower102  | CIFAR10 | CIFAR100 |
> |:------:|:--------:|:--------:|:--------:|
> |  baseline |  84.77% | 80.76% | 65.16% |
> |  $\mathcal{L}_{emb}$ |  76.78% | 84.08% | 62.97% |
> | $\mathcal{L}_{score}$ |  85.99% | 72.42% | 59.49% |
> | $\mathcal{L}_{emb+score}$ |  **86.00%** | **84.65%** | **65.96%** |
>
> The table results align and reinforce the conclusion we made in our draft: using both contrastive losses is crucial for Graph LF-CBM, as the combination of two contrastive loss terms always yields the best performances.
>
> Again, we want to reemphasize that this study aims to **investigate the importance and impacts of concept correlation inside CBMs' prediction and intervention**.
>
> [1] Coppersmith, D., & Winograd, S. (1990). "Matrix multiplication via arithmetic progressions." Journal of Symbolic Computation, 9(3), 251-280.
>
> [2] Strassen, V. (1969). "Gaussian elimination is not optimal." Numerische Mathematik, 13(4), 354-356.
>
> [3] Trefethen, L. N., & Bau, D. (1997). Numerical Linear Algebra. SIAM.
>
> [4] Gasteiger, Johannes, Aleksandar Bojchevski, and Stephan Günnemann. "Predict then propagate: Graph neural networks meet personalized pagerank." arXiv preprint arXiv:1810.05997 (2018).

---

> > ### Comment · Reviewer_8TA6 · 2024-12-02
> > **Official Comment by Reviewer 8TA6**
> >
> > 1. The original submission was difficult to understand due to unclear motivations and explanations (It seems that you have deleted the original paper). I appreciate the authors’ rebuttal, which clarified parts of the paper, including formulas, figures, and the motivation.
> >
> > 2. The title, “Graph Concept Bottleneck Models”, implies a foundational and elegant solution for extending concept bottleneck models. However, the method is built on many assumptions and complex design choices, such as:
> > 	> We hypothesize that the structure of the concept graph shall be label-agnostic.
> >
> > 	> We assume that each image can be viewed as a combination of concepts.
> >
> > These assumptions make the method specific and complicated rather than simple and general, which does not align with the expectations from the paper’s title.
> >
> > Moreover, the implementation uses many tricks and hyperparameters, which are also not well explained. For example:
> > * The motivation for DropEdge is introduced suddenly and briefly, and the dropout rate does not seem to be mentioned.
> >
> > > To prevent overfitting and over-smoothing, we employ the DropEdge Rong et al. (2020a) to randomly drop out some edges and sample a subgraph in each layer of the GNN.
> >
> > * The training requires careful tuning of parameters like $\alpha$, $\beta$, and the number of epochs, which vary across models (as noted in Appendix A.2).
> >
> > These factors suggest the model relies on complex settings rather than a simple architecture.
> >
> > 3. The graph learning part maximizes mutual information between images and their concept graphs as you mentioned. Since the image-text representations are derived from CLIP, the method essentially distills CLIP’s pretrained knowledge. This makes the performance improvement less surprising and more reliant on CLIP rather than the proposed architecture.
> >
> > Despite these issues, the paper includes interesting ideas, like using graph structures in CBMs and applying contrastive learning for concept interaction. These ideas are interesting, though they need more refinement. The authors made an effort to improve the paper during the rebuttal, which I appreciate. Based on this, I will increase the score from 5 to 6. However, my confidence in the evaluation is lower because of the heavy reliance on assumptions, tricks, and hyperparameters given in such a concise and generic paper title.

---

> ### Author Response · Authors · 2024-12-02
> **Thank you for the response**
>
> Thank you very much for your response and for increasing your score.
>
> We are glad that our rebuttals addressed some of your concerns, and we would like to elaborate more in this comment to answer your questions further.
>
> - About the complex design choice: we appreciate the comment and we agree that a foundational and general solution will be great and our method is slightly more complex than a simple solution; however we would like to argue that it is still general and foundational rather than specific. Our method can work on any scenarios that naïve CBMs are suitable for and it is even more general due to its flexibility on label-free and concept-supervised settings. Our method can also be easily extended to a simpler solution if an external graph of concept is known by removing the contrastive learning part.
> - The two hypotheses are realistic and generally applicable: the reason behind the first hypothesis about label-agnostic is further explained In lines 181 - 183. As the main theme of our paper is to discover the correlation purely existing in concept space, we intend to exclude the label-concept information which will be inevitable if we choose a label-involved learning manner. The second hypothesis about concept view can be directly derived from the CLIP models: CLIP contrastively learns a joint representation space for images and texts by viewing corresponding image-text pairs as positives. This indicates that texts are indeed augmentations of images. In our case, the concepts are textual words or sentences that describe certain attributes inside images. Therefore, it should be obvious that concept combinations are augmentation of input images.
> - About tricks and hyperparameter tuning: the number of the hyperparameters is only two and it is not hard to tune. In A.2, actually one can realize the $\alpha$ remains the same in most cases, and the $\beta$ varies due to different sensitivity to graph sparsity which is also affected by the number of concepts in different datasets. The flexibility to control the graph sparsity is useful in practice because the dense graph will hurt the interpretability when the concept set is large. As to the DropEdge operation, it is a very common trick for GNNs to avoid overfitting and oversmoothing (just like dropout for DNNs), we tested the ratio at 0 and 0.1 on one dataset (CUB) and did not see much performance variance, so finally we fixed the ratio at 0 for all other experiments.
> - For the third concern, we would like to argue that G-CBM/CEMs do not have textual CLIP embeddings, so the performance gain cannot be solely attributed to CLIP knowledge distillation. Furthermore, Using latent graphs not only benefits predictions but also helps to perform better and more robustly at intervention (section 4.5) and under attacks (section A.6). Those advantages are less relevant to the application of contrastive learning.
>
> We hope this comment can resolve some of your worries about our paper and research. Thank you again for your kind comments and suggestions.

---

### Official Review · Reviewer_r48F · 2024-11-01

**Soundness:** 2
**Presentation:** 2
**Contribution:** 3
**Rating:** 6
**Confidence:** 4

**Summary:**

This work proposes GraphCBMs, a CBM variant that aims to bypass the simplistic assumption of independent concepts via the construction of latent concept graphs.
The experimental results suggest that the proposed variant can lead to better generalization capabilities and intervention strategies.

**Strengths:**

Overcoming the commonly used assumption of independent concepts is very important to the interpretability community. Using Graph Networks is a novel and interesting approach in the context of CBMs.

**Weaknesses:**

To the best of my understanding, the proposed approach considers one concept graph per image. Even though this allows for a more flexible representation,
it may lead to some inconsistent results in terms of interpretability.

Is there a way to get a global representation of the latent structure from the localized representations?
Would it also be possible to have a variant that mainly considers a global representation that could be adjusted to individual examples?

In this context, Fig. 4 presents a concept graph for the Label-free trained CUB. Can the authors elaborate on how this graph was constructed?

Fig.4 can be improved by setting the weights of the  edges to denote the weight of the connection, or use dotted lines or other approaches.
At this point, it's not exactly easy to interpret. The authors mention some pairs, e.g., "(241) black wings with white spots - (337) white and black coloration" but there are several other connections to other nodes that are not explained, e.g., for 241, the nodes 288, 4, 245, 239.

What is the impact of the different objectives introduced by the authors? An ablation study on the impact of the different terms is imperative.

I find that some datasets for evaluating the approach are missing and specifically ImageNet and Places365. These are very common in the CBM community. Did the authors examine the behavior of the proposed approach in such a setting?

The concept prediction increase is marginal in most cases, while for CUB (in the CBM and G-CBM) case there exists a large gap. Some recent works also use other metric, such as the Jaccard index to assess the performance. Can the authors report the concept prediction performance using this metric?

There are other CBM variants that yield better results that LF-CBM and PCBMs, e.g., [1,2] (and also better than the proposed method). It would make sense to use the best performing CBM variant to assess if the emerging performance translates to such a setting.   Did the authors test any other CBMs for their approach?

In this frame of reference, the SOTA approaches are missing quite relevant CBM variants, as the ones previously mentioned.

In Table 1 and Fig. 2, which backbone is used for LF-CBM? The results don't seem to match the ones reported in the original publication.

What is the complexity of the proposed approach in terms of both training and inference? How does it compare to the corresponding baseline methods? Some wall time measurements should be enough.

Overall, I find the method to be very interesting and modelling the connections between concepts very important. However, at this time, the manuscript is missing some key datasets and comparisons with other better performing methods,


[1] Panousis et al.,  Sparse Linear Concept Discovery Models, ICCW, 2023

[2] Vandenhirtz et al, Stochastic Concept Bottleneck Models, NeurIPS 2024

**Questions:**

Please see the Weaknesses section.

---

> ### Author Response · Authors · 2024-11-23
> **Response to r48F (1)**
>
> Dear Reviewer r48F,
>
> We appreciate the effort you have put into reviewing our paper. We believe there are some misunderstandings, and we would like to address your concerns in this rebuttal response as best as we can.
>
> “*The proposed approach considers one concept graph per image.*”
> 1. We believe there are some misunderstandings here and please let us clarify. Our
> proposed latent graph structure is for dataset level rather than instance level, i.e., images in the same dataset share the same latent graph structure. We also explicitly wrote this notion inside our submission, in lines 184-186: *Although the graph structure $\mathcal{A}$ is unified at the **dataset level** and **fixed for all images**, the concept scores decide a different subset of concepts that are activated by each image.* The rationale behind setting a latent graph at the dataset level is that the concept set is also defined at the dataset level. Since the latent graph aims to discover the relations and dependency among concepts, a dataset-level concept set is reasonable to have a dataset-level concept graph.
>
> “*Is there a way to get a global representation of the latent structure from the localized representations? Would it also be possible to have a variant that mainly considers a global representation that could be adjusted to individual examples?*”
>
> 2. To avoid confusion and misunderstanding, we want first to define the meaning of global and local representations. Given the latent graph, the global representation is the readout value of all the node representations inside the graph, and the local representation will be the same readout function acting on a subgraph.
>
> To answer your first question: because different images will activate different concepts (by measuring the similarity between the image and each concept), the activated concepts can be regarded as localized representations. Recall that the node features inside the latent graph are the concept embeddings from the text encoder, the global representation of the latent graph for each image will be distinct due to different activated subgraphs. In this case, the global representation is highly correlated to the localized activated subgraphs or representations.
>
> For your second concern, we want to clarify that we indeed already consider **all concept information** for prediction. As described in our draft (lines 186-187): *That means we can have a graph with different node activations for each image. Thus, the graph representation should be an augmented view of the image representation $z_{v_{i}}$.* Along with the illustrations in lines 184-186 and our definition above, we choose the global representation and all concept information for making the prediction, and global representation and concept information will be adjusted to individual examples. We will make this point more clear in the revision to avoid confusion.
>
> “*In this context, Fig. 4 presents a concept graph for the Label-free trained CUB. Can the authors elaborate on how this graph was constructed?*”
>
> 3. As we explained in Line 160, the latent concept graph is learned by defining a learnable non-negative adjacency matrix A. The generation of the graph is described in Sec 3.2.1, through the optimization of a contrastive learning objective. Let us briefly clarify the process below for your better understanding.
>
> First, we made three assumptions:
> - i. The concept graph shall be label-agnostic, and we do not want much label information to get involved.
> - iii. The combination of the activated concepts can be an augmented view of the input image, and the mutual information between different views of an image should be maximized.
> - iii. Concept score vectors are some hidden representations of input images.
>
> To the hypotheses above, we choose to use contrastive learning to guide the latent graph generation.
>
> In lines 230-233: *$\mathcal{L}{emb}$ and $\mathcal{L}{score}$ execute contrastive learning at the different layers, which controls the quality of graph structures in various aspects. We combine the two contrastive learning objectives as well as a regularization term to control the sparsity of the learned graph and obtain the final contrastive learning loss.* Since the inputs $z_{g_{i}}$ and $c'_{i}$ to $\mathcal{L}{emb}$ and $\mathcal{L}{score}$ are varied from image to image, the contrastive objectives will help to cluster those activated concepts and act as the supervision for the latent graph structuring. And the $L-1$ normalization will swipe out those less significant relations to control the sparsity. By looking at Fig 4, we witness the reasonable relations captured by the latent graph, so we can conclude the effectiveness of our training process afterward.

---

> ### Author Response · Authors · 2024-11-23
> **Response to r48F (2)**
>
> “*Fig.4 can be improved by setting the weights of the edges to denote the weight of the connection, or use dotted lines or other approaches.*”
>
> 4. We appreciate your valuable advice and we will make this change in the revision. For your concern about the concept pairs, we only excerpt a subset of the whole concept pair sets (as stated in lines 465 - 466). Fig 4 presents a rough visualization of the latent graph and gives evidence of the latent graph's effectiveness. We will include the original concept pair set in the appendix in the coming revision.
>
> “*What is the impact of the different objectives introduced by the authors? An ablation study on the impact of the different terms is imperative.*”
>
> 5. We did provide ablation studies for the impact of contrastive objectives in Appendix A.9 and we also mention this reference in the main context in lines 317-318: *We provide a detailed analysis of the importance of contrastive terms in Appendix A.9.* In A.9, we have empirically shown that the two contrastive terms are essential for label-free setting CBM to have the optimal performance, having just one of them can only yield suboptimal or sometimes degraded performances.
>
> “*ImageNet and Places365.*”
>
> 6.
> | Model |  Places365   |  ImageNet  |
> |:------:|:--------:|----------|
> |  PCBM  | 55.24%  | 77.48% |
> | G-CBM | **55.31%**  | **78.47%** |
>
> Above are the experiment results of PCBM and G-PCBM on Places365 and ImageNet. Considering the limited discussion time, we only train all models for 10 epochs on both datasets for 3 different random seeds, and we perform submodular filtering to select 200 concepts for each dataset. The hyperparameter settings are in the table below.
>
>  | Dataset |  Learning Rate   |  $\alpha$  |    $\beta$    |
> |:------:|:--------:|:--------:|:--------:|
> |  Places365  | 0.01  | 0.1 | 0.15 |
> | ImageNet | 0.01  | 0.1 | 0.17 |
>
> Both are trained with Adam Optimizer with a batch size of 1024.
>
> To further show the benefit of using latent graphs, we also investigate the intervention effects and we offer a case study on ImageNet as an illustration for intervention enhancement at different ratios in the following table.
>
> | Model |  0.25   |  0.5  | 0.75 | 1.0 |
> |:------:|:--------:|:--------:|:--------:|:--------:|
> |  PCBM  | 86.42%  | 89.53% | 92.05% | 94.61% |
> | G-CBM | **91.08%**  | **92.80%** | **93.89%** | **94.92%** |
>
> With latent graphs, intervention can be more effective, especially with a smaller intervention ratio and the latent graph can more substantially improve model intervention performance.
>
> “*Jaccard index*”
>
> 7. CBM is well-known for its interpretability which is not only shown in its ability to make label and concept predictions but also in its intervention behavior. The essential benefit of G-CBM is to capture the correlation of concepts and encourage the concept score to be updated before making the final label prediction. So, it is intuitive that the concept prediction accuracy will not be impacted too much if we take the same backbone for CBMs and G-CBMs; instead, it will be more effective in the final label prediction as well as the intervention. Furthermore, our model is not limited to the supervised concept setting, and it is flexible to be adapted to the label-free setting where no concepts are labeled and concept accuracy cannot even be evaluated. We also want to emphasize that in Table 2, compared to CEMs, one of the state-of-the-art CBM variants, our proposed architectures G-CBM and G-CEM improve the prediction accuracy to match or even surpass CEMs results while shrinking the gap of concept prediction.
>
> For the Jaccard index evaluation, please refer to the table below
>
> | Model |   CUB   |  AwA2  | ChestXpert |
> |:------:|:--------:|----------|:---------:|
> |  CBM  | 37.29%  | 93.62% |   52.10%   |
> | G-CBM | 35.13%  | 92.84% |   55.10%   |
>
> Also, we would like to claim that the latent graph is not designed to increase concept accuracy. The latent graph intends to find the hidden correlation among concepts and activates those hidden concepts to help label/target prediction. Therefore, it can help to activate some concepts that are not directly present in the input image. For example, by taking different views, images describing the same labeled object can give different visual concepts, and latent graphs can find those inhibited concepts to help label prediction with the concept accuracy loss of answering those hidden concepts.
>
> To prevent misunderstanding and make our claims clearer, we will add a sentence to our introduction such that our models can also mitigate the accuracy-interpretability trade-off, which should be a bonus. We want to once again underline that the main goal of our paper and studies is to **make CBMs realize concept correlations**, and we hence proposed Graph CBMs to model the latent concept graph for capturing concept dependencies and relations.

---

> ### Author Response · Authors · 2024-11-23
> **Response to r48F (3)**
>
> “*CDM and S-CBM*”
>
> 8. Thank you for pointing out two new CBM variants. We have also reproduced CDM[2] and S-CBM[1] on the CUB datasets. We follow the paper instructions, GitHub references, and the same hyperparameter settings proposed in [1,2], training models on 10 different random seeds. The original [1] uses RN18 as the backbone and [2] selects CLIP-RN50 and CLIP-ViTB/16 as backbones.  For a fair comparison, we fix all backbone models to CLIP-ViT-B/16.
>
> |  Model |   CUB  |
> |:------:|:--------:|
> |  CDM |  73.53% |
> | G-CDM | **77.16%** |
> | G-PCBM | 77.14% |
>
> In the table above, we can see that the CDM falls short in comparison to G-PCBM, and we can adapt our proposed latent graph idea on top of CDM, which further boosts the label prediction accuracy to match with G-PCBM.  The CDM uses a concept presence indicator which is modeled by a Bernoulli distribution to sparsify the concept scores. Such sparse concept scores will lead to some information loss and harm the label prediction. On the other hand, latent graphs assist the model in recovering those missed concept information through message passing, and the results also verify this point. We have studied a similar behavior in Appendix A.5, in which we perform different attacks on concepts, and shown that models with latent graphs are more robust under attacks.
>
> |  Model |   Label (Acc)  | Concept (Jaccard) | Training Speed |
> |:------:|:--------:|:--------:|:--------:|
> |  S-CBM |  74.75% | **35.73%** | 11.35s/it |
> | G-CBM |  **80.03%** | 35.13% | **6.12it/s** |
>
> S-CBM shares a similar motivation as ours to study the concept dependency among concepts. S-CBM takes an approach on an explicit, distributional parameterization method while we choose graph structure learning to model a latent concept graph. In the above table, S-CBM can only obtain a label accuracy lower than 75% on the CUB dataset, and G-CBM can achieve over 80% accuracy. In the concept prediction, G-CBM also matches up with S-CBM. Furthermore, S-CBM needs to train 300 epochs with 11.35s taken at each iteration, and G-CBM is more efficient as it only requires 100 epochs with 6.12s taken at each iteration.
>
> By comparing with other CBM variants, we have further proved the effectiveness of latent graphs, and we appreciated your valuable advice for more SOTA comparison experiments. We will include those experiments in our appendix and cite those two papers [1,2] in our paper revision.
>
> “*In this frame of reference, the SOTA approaches are missing quite relevant CBM variants, as the ones previously mentioned.*”
>
> 9. Please see our response 8. We want to further clarify and discuss that the motivations between our Graph CBMs and [2] are essentially different. [2] aims to continue to sparsify the concept sets, and our focus is to discover the concept correlations. [1] shares a similar motivation as ours but chooses a different methodology.
>
> Furthermore, our proposed latent graph is more versatile compared to CDM and S-CBM. CDM mainly works for label-free settings, while S-CBM is designed for datasets with concept annotations. The concept latent graph is capable of capturing the intrinsic relations among concepts regardless of the training settings and making positive impacts on predictions and interventions.
>
> “*LF-CBM Mismatch*”
>
> 10. We followed the same backbones as LF-CBM did in [3], i.e., we used CLIP(RN50) image encoder as the backbone for CIFAR, Flower102, HAM10000, and ResNet-18 trained on CUB from imgclsmob for CUB. We also mention this in lines 241 - 243. The reason for the different performance in the CIFAR10 dataset is that we use a smaller and different concept set (only selecting 30 concepts). In the following table, we reproduce the experiment on the CIFAR10 by using the same concept set in [3] and trained through 10 different random seeds.
>
> |  Model |   CIFAR10  |
> |:------:|:------:|
> |  LF-CBM |  86.40% |
> | G-LF-CBM | **86.54%** |
>
> The performance on their large concept set (with 100+ concepts) is only marginally better, however as we showed in the paper, G-LF-CBM is much more efficient with fewer concepts, and the latent graph helps more during interventions.
>
> “*Efficiency of G-CBM*”
>
> 11. Yes, we agree that we should involve some time measurement to indicate the efficiency of our proposal. We will add the following table in the appendix.
>
> |  Model |   Train  | Test     |
> |:------:|:--------:|----------|
> |  PCBM  |  5.7it/s | 2.09it/s |
> | G-PCBM | 9.43it/s | 2.12it/s |
> |----------|----------|----------|
> |  CBM  |  6.12it/s | 2.90it/s |
> | G-CBM | 7.89it/s | 2.57it/s |
>
> [1] Vandenhirtz, Moritz, et al. "Stochastic Concept Bottleneck Models." arXiv preprint arXiv:2406.19272 (2024).
>
> [2] Panousis, Konstantinos Panagiotis, Dino Ienco, and Diego Marcos. "Sparse linear concept discovery models." Proceedings of the IEEE/CVF International Conference on Computer Vision. 2023.
>
> [3] Oikarinen, Tuomas, et al. "Label-free concept bottleneck models." arXiv preprint arXiv:2304.06129 (2023).

---

> > ### Comment · Reviewer_r48F · 2024-11-26
> >
> > I thank the authors for their thorough response. Most of my concerns have been addressed and I'm content with the additional clarifications and experiments.
> >
> > I would propose that the authors incorporate all the changes and address all the points raised by the other reviewers it their manuscript since it will help with the readability and impact of the work.

---

> ### Author Response · Authors · 2024-11-27
> **Thank Response**
>
> Dear reviewer r48F,
>
> We are glad that our responses addressed your concerns and grateful for your advice which will help us significantly improve the quality and clarity of the paper. Your insights have been incredibly valuable in helping us improve the paper. We have revised the draft according to your suggestions and other reviewers' comments. There are more elaborations highlighted in different colors in the draft to ameliorate the clarity and readability of the context. If you have any other concerns, we will be more than willing to assist you in answering questions.
>
> Many thanks!

---

### Official Review · Reviewer_J5ky · 2024-11-08

**Soundness:** 3
**Presentation:** 3
**Contribution:** 3
**Rating:** 6
**Confidence:** 2

**Summary:**

This paper proposes Graph Concept Bottleneck Models (Graph CBMs) to address the limitations of existing CBMs. The authors identify that while CBMs provide interpretable models by mapping inputs to a concept score space, the assumption of conditional independence among concepts is often violated in practice. To capture the intrinsic structure and relationships within the concept space, Graph CBMs construct latent concept graphs and combine them with CBMs. The empirical results demonstrate that Graph CBMs outperform standard CBMs on image classification tasks, can utilize the concept graphs for more effective interventions, and are robust across different training and architecture settings. Overall, this work presents a significant advancement in improving the performance and interpretability of CBMs by explicitly modeling the hidden interactions within the concept space.

**Strengths:**

1. This paper is the first to propose CBM that utilizes a graph structure to facilitate interaction between concepts. This advancement represents a significant improvement over previous models in understanding the dependencies among concepts. Such an introduction enhances the model's ability to capture complex relationships between concepts effectively.

2. The research demonstrates the superior performance of Graph CBM across various tasks and different CBM architectures through detailed experiments. Furthermore, Graph CBM allows for human intervention, showcasing strong scalability and providing greater flexibility for real-world applications.

3. The graph structure of Graph CBM provides improved interpretability in the model's decision-making processes. This enables users to gain a clearer understanding of how the model operates and the rationale behind its decisions, thereby increasing the transparency of the model in practical applications.

**Weaknesses:**

1. While the incorporation of a combined contrastive loss from the score space and the embedding space is an innovative design, the paper may lack a detailed explanation of the motivation behind this choice. This insuff can hinder the readers' ability to fully understand the theoretical foundation and practicality of this approach. Furthermore, without supporting ablation studies, readers may struggle to assess the specific impact of this design on the model's performance.


2. Although Graph CBM claims to effectively learn dependencies between concepts through its graph structure, the paper may not provide systematic experimental results to convincingly validate this assertion. This raises questions about the source of performance improvements in Graph CBM, as any enhancements in specific cases may not be wholly attributed to the introduction of the graph structure.

3. Despite the enhanced capability of Graph CBM in learning dependencies, the absence of specific visual examples may limit readers' understanding of this improvement.

**Questions:**

1. Why was a combined contrastive loss from the score space and embedding space adopted during the training of Graph CBM? What is the motivation behind this design? Can ablation studies be conducted to provide more intuitive results demonstrating the specific impact of this loss formulation on model performance?

2. Does the performance improvement of Graph CBM over traditional CBM stem from its graph structure effectively learning the dependencies between concepts? Is there specific experimental evidence to support this claim?

3. Related to the second question, given that Graph CBM performs better than traditional CBM in learning dependency relationships, can specific visual examples be provided to clearly demonstrate this enhanced performance?

---

> ### Author Response · Authors · 2024-11-23
> **Response to J5ky (1)**
>
> Dear reviewer J5ky,
>
> We are grateful for the time and effort you spent reviewing our paper, and we hope our rebuttal response can answer your questions.
>
> “*Reasons and motivations of two contrastive losses and intuitive ablation studies.*”
>
> 1. In the CBM setting, the image encoder and the text encoder are usually frozen. Therefore, only a small number of learnable projection heads will be activated during the training, and the rationale behind this is to make CBM more lightweight in terms of training parameters and promote more direct interpretation. However, such an insufficient model depth will lead to sub-optima of contrastive learning. [1], [2], and [3] all revealed the impact of model depth and model size on contrastive learning performance in different extents. Thus, to overcome the drawback of a few training parameters, we proposed to contrast the image and the latent graph (concepts as nodes) at different granularity levels. In the main context (lines 220-221), we also provide some references for readers to look up the motivation of multi-contrastive learning in general.
>
> The contrastive losses also stem from these fundamental assumptions:
>
> - i. The concept-latent graph should be label-agnostic.
> - ii. A combination of concepts can be viewed as an augmentation of the input image.
> - iii. Concept score vectors are some hidden representations of input images.
>
> Thus, we adopt contrastive learning to supervise latent graph generation, and the two contrastive losses also act on different aspects of the latent graph generation.
> $\mathcal{L}_{emb}$ directly controls the alignment and uniformity between text/concept embeddings and image embeddings. The latent graph plays a role more on the operation side and focuses more on the impact of message passing.
>
> $\mathcal{L}_{score}$ on the other hand mainly aims to make the concept score vectors better hidden representations of input images. The latent graph will work more like a hidden feature rather than an operation.
>
> We also empirically show in our paper (Appendix 9) that only considering a single contrastive term will result in suboptimal performance, so combining two contrastive losses will be necessary in our settings. Thus, only having the latent graph as a message-passing operation or a hidden feature falls short of making the best results.
>
>
> “*Does the performance improvement of Graph CBM over traditional CBM stem from its graph structure effectively learning the dependencies between concepts? Is there specific experimental evidence to support this claim?*”
>
> 2. We would like to elaborate on the essential and the only difference between our proposed model architecture and chosen baselines is the latent graph. We will update concept score values through message passing with the learned latent graph, and that is the only new operation in the whole training and inference process. During training, we rigorously keep the image feature out of label prediction, i.e., the model can only make label prediction inside concept space strictly. In this manner, the performance enhancement can be solely attributed to the latent graph. Notice that contrastive learning only contrasts the input image and the latent graph (because of its self-supervised manner), in which we prevent label information leakage. In section 4.6, we can find out that the latent graph effectively discovers reasonable correlations among concepts. So we can conclude:
>
> - A. The only new operation in our proposed method is to use latent graph message passing, and our method can match up and surpass multiple (sota) baselines in label predictions and human interventions.
> - B. The latent graph successfully captures reasonable concept relations.
>
> By considering A and B, we can deduce that knowing concept correlation can boost CBM performance in prediction and intervention.

---

> ### Author Response · Authors · 2024-11-23
> **Response to J5ky (2)**
>
> “*Related to the second question, given that Graph CBM performs better than traditional CBM in learning dependency relationships, can specific visual examples be provided to clearly demonstrate this enhanced performance?*”
>
> 3. We would argue that Figure 5 gives a good visualization as it shows that concepts belonging to the same label class are more likely to be connected inside the latent graph, and the Figure 8, the salient graph further filters out redundant edges, and the remaining edges even more possible to connect concepts describe the same label. However, we would be thankful if you could give us a more detailed and clear instruction about what kind of visualization you think will better demonstrate the enhanced performance.
>
> Besides those answers, we want to claim that intervention on latent graphs is highly non-trivial. The difficulties mainly come from 2 reasons.
>
> - i. If the concept is not defined at the label level, i.e., the same labeled images have different concept distribution. In this case, finding other reasonable connections or edges can be time-consuming as we need to manually look at those images.
> - ii. After we know which concepts need to be present, assigning node attributes also creates an enormous obstacle.
>
> Based on the finding above, one expects that replacing the learned latent graph with any random latent graph will significantly affect model performance negatively. Consequently, we decide to do a post-hoc analysis to interpret the meaning and function of the latent graph.
>
> [1] Chen, Ting, et al. "A simple framework for contrastive learning of visual representations." International conference on machine learning. PMLR, 2020.
>
> [2] Wang, Tongzhou, and Phillip Isola. "Understanding contrastive representation learning through alignment and uniformity on the hypersphere." International conference on machine learning. PMLR, 2020.
>
> [3] Tan, Mingxing, and Quoc Le. "Efficientnet: Rethinking model scaling for convolutional neural networks." International conference on machine learning. PMLR, 2019.

---

### Author Response · Authors · 2024-11-27
**General Response For The Paper Revision**

Dear reviewers and ACs,

We appreciate the reviewers' valuable advice and sincerely believe that insightful feedback will help us significantly improve the quality and clarity of the paper. We have revised our draft according to every reviewer's comments and provided detailed answers in our response to each reviewer. This comment aims to briefly summarize the changes we made to our paper:

1. We added more sentences highlighted in orange color to ameliorate the clarity and readability. We primarily focused on the motivation behind contrastive learning by explicitly stating our essential assumptions: 1) label-agnostic learning pattern for latent graphs and 2) activated concepts as image augmentations.
2. We offered our new experiments on large-scale datasets like Place365 and ImageNet in A.4 and mentioned the new add-on results in the main content.
3. We also compared our proposed method to more SOTAs (S-CBM [1] and CDM [2]) in Figure 2. In A.7, we test our latent graph idea on the CDM backbone to reinforce our claim that latent graphs are orthogonal to SOTAs' approaches and can be applied in any CBM backbone.
4. We expanded Table 9 to 4 datasets: CUB, HAM10000, Flower102, and CIFAR100. The new experiments aligned with our previous conclusion about the importance of having two contrastive losses.
5. In section 4.3, we emphasized that the latent graph preferentially contributed to label prediction as it can activate inhibited concepts in the image to gather more useful information.
6. In lines 399-404, we pointed out the unique latent graph's supremacy: versatility. Unlike most current SOTA approaches, one can adapt the latent graph idea to any training setting, i.e., label-free and concept supervision.
7. In Figure 1, we replace the original sketchy graph with dense and sparse learned latent graphs to better present readers with a sense of the graph evolution throughout the training process, i.e., from a chaotic and dense graph to an informative and sparse graph.
8. We attached the overview of the whole latent graph along with the connectivities in Figure 13.
9. We fixed some typos in the draft and added a time measurement table in A.2.

We sincerely hope that the responses and the revised paper assist in your re-evaluation of our paper. We are also more than willing to help you with other questions on our paper.

Many thanks!

[1] Vandenhirtz, Moritz, et al. "Stochastic Concept Bottleneck Models." arXiv preprint arXiv:2406.19272 (2024).

[2] Panousis, Konstantinos Panagiotis, Dino Ienco, and Diego Marcos. "Sparse linear concept discovery models." Proceedings of the IEEE/CVF International Conference on Computer Vision. 2023.

---

### Meta-Review · Area_Chair_YWc6 · 2024-12-19

**Metareview:**

The paper titled "Graph Concept Bottleneck Models" introduces Graph CBMs, an extension of traditional Concept Bottleneck Models (CBMs) that incorporates latent concept graphs to capture dependencies among concepts. The authors argue that existing CBMs assume conditional independence among concepts, which is often unrealistic, and propose Graph CBMs to model intrinsic concept relationships. Empirical evaluations on multiple image classification datasets demonstrate that Graph CBMs outperform standard CBMs in terms of prediction accuracy, intervention effectiveness, and robustness across different training and architectural settings. Additionally, the paper claims enhanced interpretability through the structured concept graphs.

Despite its innovative approach, the paper presents several significant weaknesses that hinder its acceptance. The motivation behind the combined contrastive losses and the choice of graph propagation mechanisms lacks clarity and sufficient theoretical justification. Comparisons with existing state-of-the-art CBM variants, such as EfficientDM and S-CBM, are either missing or inadequately addressed, raising concerns about the method's relative effectiveness and generalizability. Furthermore, the experimental setup suffers from limited ablation studies and insufficient evaluations on standard benchmarks like ImageNet and Places365, which diminishes the reliability of the claimed performance improvements. The complexity of the proposed method, reliance on multiple hyperparameters, and lack of comprehensive explanations for key design choices further obscure the paper's contributions and limit its practical applicability. These unresolved issues and methodological shortcomings lead to the recommendation to reject the paper in its current form.

**Additional Comments On Reviewer Discussion:**

During the rebuttal phase, the authors addressed several key concerns raised by the reviewers. Reviewer J5ky requested more ablation studies on contrastive losses and clarification on the source of performance gains. The authors responded by adding relevant experiments in the appendix and attributing performance improvements to the introduction of latent concept graphs. Reviewer r48F misunderstood the concept graph construction, initially believing it to be image-specific rather than dataset-level. The authors clarified that the latent graph is shared across the dataset and provided additional experiments on larger datasets like Place365 and ImageNet to demonstrate the method's scalability and effectiveness. Reviewer 8TA6 criticized the paper's unclear motivations, complex design choices, and insufficient experimental validation. While the authors elaborated on their design choices and included additional experimental results, the explanations remained insufficient to fully alleviate the reviewer's concerns. Reviewer TaXb highlighted the lack of neuron interpretability and the method's reliance on numerous hyperparameters, which the authors addressed by clarifying the focus on concept-level interpretability and providing further experimental data. Despite these efforts, key issues regarding the paper's clarity, methodological robustness, and comprehensive empirical validation remained unresolved, ultimately leading to the decision to reject the submission.

---

### Decision · Program_Chairs · 2025-01-22

Reject